# Heparan sulfate-dependent RAGE oligomerization is indispensable for pathophysiological functions of RAGE

**Miaomiao Li[1], Chih Yean Ong[1], Christophe J Langouët-Astrié[2], Lisi Tan[1,3], Ashwni Verma[4], Yimu Yang[2], Xiaoxiao Zhang[1], Dhaval K Shah[4], Eric P Schmidt[2], Ding Xu[1]***

[1]Department of Oral Biology, University at Buffalo, State University of New York, Buffalo, United States; [2]Division of Pulmonary Sciences and Critical Care Medicine, Department of Medicine, University of Colorado Anschutz Medical Campus, Aurora, United States; [3]Department of Periodontics, School of Stomatology, China Medical University, Shenyang, China; [4]Department of Pharmaceutical Sciences, School of Pharmacy and Pharmaceutical Sciences, University at Buffalo, State University of New York, Buffalo, United States

**Abstract** RAGE, a druggable inflammatory receptor, is known to function as an oligomer but the exact oligomerization mechanism remains poorly understood. Previously we have shown that heparan sulfate (HS) plays an active role in RAGE oligomerization. To understand the physiological significance of HS-induced RAGE oligomerization in vivo, we generated RAGE knock-in mice (*Ager*^AHA/AHA) by introducing point mutations to specifically disrupt HS-RAGE interaction. The RAGE mutant demonstrated normal ligand-binding but impaired capacity of HS-binding and oligomerization. Remarkably, *Ager*^AHA/AHA mice phenocopied *Ager*^−/− mice in two different pathophysiological processes, namely bone remodeling and neutrophil-mediated liver injury, which demonstrates that HS-induced RAGE oligomerization is essential for RAGE signaling. Our findings suggest that it should be possible to block RAGE signaling by inhibiting HS-RAGE interaction. To test this, we generated a monoclonal antibody that targets the HS-binding site of RAGE. This antibody blocks RAGE signaling in vitro and in vivo, recapitulating the phenotype of *Ager*^AHA/AHA mice. By inhibiting HS-RAGE interaction genetically and pharmacologically, our work validated an alternative strategy to antagonize RAGE. Finally, we have performed RNA-seq analysis of neutrophils and lungs and found that while *Ager*^−/− mice had a broad alteration of transcriptome in both tissues compared to wild-type mice, the changes of transcriptome in *Ager*^AHA/AHA mice were much more restricted. This unexpected finding suggests that by preserving the expression of RAGE protein (in a dominant-negative form), *Ager*^AHA/AHA mouse might represent a cleaner genetic model to study physiological roles of RAGE in vivo compared to *Ager*^−/− mice.

*For correspondence:
dingxu@buffalo.edu

## Editor's evaluation

The Receptor for Advanced Glycation End-products (or RAGE) has garnered great interest over the past 20 years for its role in the complications of diabetes mellitus and in Alzheimer's disease, atherosclerosis and other inflammatory disorders. RAGE has several ligands. This paper explores the role of heparan sulfate in the oligomerization of RAGE and the role of oligomerization in vivo function using mouse knockout models. The authors report that knock-in mice, with RAGE is mutated at sites responsible for heparan sulfate binding and oligomerization, phenocopy RAGE knockout mice. They further validate the idea that this knock-in mouse, which preserves expression

and binding of RAGE, may be a valuable model for studying this important molecule in disease pathogenesis.

## Introduction

Receptor for Advanced Glycation End-Products (*Ager* gene) encodes RAGE, a transmembrane protein belonging to the immunoglobulin (Ig) superfamily of receptors. RAGE has attracted tremendous interest in the last two decades given its involvement in many disease conditions including diabetes, Alzheimer's, atherosclerosis, cancer, and many other inflammatory disorders (*Hudson and Lippman, 2018*; *Yan et al., 2010*). RAGE binds a wide array of ligands including advanced glycation end products, amyloid β-peptide, S100 family proteins, and high mobility group Box 1 (HMGB1). In most pathological conditions, diseased tissues show a dramatically enhanced expression and activation of RAGE. Ablation of RAGE signaling in these conditions, either by neutralizing RAGE ligands or by using RAGE knockout mice, has resulted in greatly improved outcome in murine models, which strongly suggests that RAGE can be pursued as a promising therapeutic target (*Arnold et al., 2020*; *Harja et al., 2008*; *Hofmann et al., 1999*; *Huebener et al., 2015*; *Park et al., 1998*; *Taguchi et al., 2000*).

While many pathological roles of RAGE have been well established, the homeostatic role of RAGE remained a mystery until 2006, when it was found that RAGE plays an essential role in regulating osteoclast differentiation and maturation (*Zhou et al., 2006*). While unchallenged RAGE null mice display normal homeostasis in all other aspects, they demonstrated increased bone mineral density. Mechanistically, it was revealed that HMGB1 secreted by preosteoclasts activates RAGE in an autocrine fashion, and HMGB1-RAGE interaction plays a critical role in establishing the sealing zone, a structural feature of osteoclasts essential for bone resorption (*Zhou et al., 2008*).

Recently, HMGB1-RAGE signaling has been shown to also play a key role in drug-induced liver injury (*Huebener et al., 2015*). Following acetaminophen (APAP) overdose, hepatocytes undergoing necrosis release HMGB1 as a danger signal to promote tissue repair. However, when extensive tissue necrosis occurs, the exaggerated inflammation, mediated by HMGB1-driven, RAGE-dependent neutrophil infiltration, often leads to secondary liver damage with worsened disease outcome. In a mouse of APAP-induced liver injury, we and others demonstrated that *Ager*$^{-/-}$ mice were greatly protected from liver damage chiefly due to suppression of neutrophil infiltration (*Arnold et al., 2020*; *Huebener et al., 2015*).

Like many cell surface receptors, RAGE signaling depends on receptor oligomerization (*Xie et al., 2008*; *Popa et al., 2014*; *Zong et al., 2010*). However, the exact mechanism by which RAGE oligomerizes remains controversial (*Moysa et al., 2019*). Historically, RAGE has been hypothesized to undergo ligand-dependent oligomerization (*Xue et al., 2016*). We have proposed an alternative hypothesis: that prior to ligand binding, RAGE preassembles into an oligomeric complex by interacting with heparan sulfate (HS), a negatively charged polysaccharide widely expressed at the cell surface. This alternative hypothesis is supported by in-depth structural analyses, where we showed by both crystal and solution structures that HS oligosaccharides induce RAGE to form a stable hexamer in the absence of ligand (*Xu et al., 2013*). Furthermore, in the same study, we showed that in endothelial cells, activation of RAGE signaling by all tested RAGE ligands, including HMGB1, S100A8/A9, S100A12, S100B, and AGE, depended upon the presence of HS at the cell surface. Based upon these findings, we propose that HS functions as a main driver of RAGE oligomerization, assembling RAGE into an oligomeric complex at cell surface without activating it. Only this oligomeric complex can serve as a functional receptor, which can then be activated by RAGE ligands to initiate signaling. As virtually all mammalian cells express HS, HS-dependent RAGE oligomerization is likely a universal requirement for RAGE signaling regardless of cell types and ligands involved.

To investigate the physiological significance of HS-RAGE interaction and HS-dependent RAGE oligomerization, we generated a novel RAGE knock-in (*Ager*$^{AHA/AHA}$) mouse model by introducing point mutations into residues that are involved in HS binding. Remarkably, when bone morphometric indexes and susceptibility to APAP-induced liver injury were examined, we found that the *Ager*$^{AHA/AHA}$ mice phenocopy *Ager*$^{-/-}$ mice precisely. Our findings strongly suggest that HS-RAGE interaction is required for normal function of RAGE in multiple cellular systems and that targeting HS-RAGE interaction could represent a new opportunity to curb RAGE activation. By developing a monoclonal antibody (mAb) that specifically targets part of the HS-binding site of RAGE, we demonstrated that

this novel targeting strategy was indeed effective in inhibiting osteoclastogenesis and APAP-induced liver injury. Finally, we discovered that compared to *Ager*[−/−] mice, alterations in global transcriptome were much more restricted in *Ager*[AHA/AHA] mice, which suggest that *Ager*[AHA/AHA] mice might be a better murine model to study RAGE-dependent biological processes.

## Results

### Characterization of RAGE-AHA mutant and generation of *Ager*[AHA/AHA] knock-in mice

RAGE is a multidomain protein consisting of three extracellular immunoglobulin domains (V-C1-C2, also known as sRAGE), a single transmembrane domain, and a short cytoplasmic tail (*Dattilo et al., 2007*; *Koch et al., 2010*). We previously reported that HS can induce RAGE to form a hexamer (trimer of dimers) with 1:2 stoichiometry, and that the interaction involves five basic residues in the V domain and two residues (R216 and R218) in the C1 domain (*Figure 1A*; *Xu et al., 2013*). Here, we prepared a triple mutant of mouse RAGE VC1 domain bearing R216A, R217H, and R218A mutations (mVC1-AHA). As expected, the binding capacity of mVC1-AHA to the heparin Sepharose column was reduced significantly compared with wild-type mouse RAGE VC1 domain (WT-mVC1) (*Figure 1B*). Furthermore, mVC1-AHA was unable to form stable hexamer in the presence of HS dodecasaccharides (*Figure 1C*). Importantly, mVC1-AHA maintains WT-like binding affinity to RAGE ligand HMGB1 (apparent binding affinity 9.9 nM vs. 11.7 nM, *Figure 1D*), which is expected because binding of RAGE ligands predominantly occurs at the V domain. To generate a murine model to specifically disrupt HS-RAGE interaction, we introduced R216A-R217H-R218A triple mutations into the *ager* (synonymous with *rage*) locus by CRISPR-Cas9 mediated homologous recombination (*Figure 1E*). *Ager*[AHA/AHA] mice were born with normal litter size and grew normally without gross morphological abnormalities. The RAGE expression level in *Ager*[AHA/AHA] mice lungs was similar to WT mice, which suggested that the mutation did not negatively affect the expression of RAGE (*Figure 1F*).

### *Ager*[AHA/AHA] and *Ager*[AHA/+] mice develop an osteopetrotic phenotype

As RAGE was reported to be associated with osteoclast maturation (*Zhou et al., 2006*), we first examined the bone phenotype of *Ager*[AHA/AHA] mice. Micro-computed tomography (micro-CT) analysis of the femoral trabecular bone morphology and microarchitecture showed that 10-week-old male *Ager*[AHA/AHA] mice exhibited increased trabecular bone mass (*Figure 2A*) and displayed a 49% increase in bone volume over tissue volume (BV/TV) (*Figure 2B*), 17% increase in trabecular thickness (Tb.Th), and 15% decrease in trabecular separation (Tb.Sp) (*Figure 2—figure supplement 1*). In addition, these *Ager*[AHA/AHA] mice also displayed a significant increase in the cortical bone thickness (from 0.182 to 0.205 mm) (*Figure 2C*). Similar phenotype was also observed in 10-week-old female *Ager*[AHA/AHA] mice, which displayed a 70% increase in bone volume compared to WT mice (*Figure 2—figure supplement 2*). Notably, all these parameters are highly similar to those of 10-week-old *Ager*[−/−] mice (*Figure 2A-C* and *Figure 2—figure supplements 1–2*). Interestingly, when they were 4 weeks old, the *Ager*[AHA/AHA] mice had normal bone mass, while *Ager*[−/−] mice had slightly lower bone mass compared to WT mice. By 4 months old, while the bone mass of *Ager*[AHA/AHA] mice was 26% higher than WT mice, the bone mass of *Ager*[−/−] mice was significantly higher (75% over WT). Taken together, these data indicate that the interaction between HS and RAGE, and HS-induced oligomer formation, are indispensable for RAGE to function in bone remodeling.

To understand the potential contribution of haploinsufficiency and RAGE gene dosage in bone phenotype, we examined both strains of heterozygous mice. To our surprise, male *Ager*[AHA/+] mice already manifested significant increase in BV/TV (30%) and in cortical bone thickness (from 0.182 to 0.198 mm) (*Figure 2A–C*). In contrast, although an upward trend was observed in male *Ager*[+/−] mice, they did not display significant increase in either parameter (*Figure 2B–C*). However, 10-week-old female *Ager*[+/−] mice displayed a significant increase in BV/TV (49%) which was almost identical to the increase observed in female *Ager*[AHA/+] mice (51% increase) (*Figure 2—figure supplement 2*). Of note, while the BV/TV value was significantly different between *Ager*[+/−] and *Ager*[−/−] mice in both male and female mice, it was not significantly different between *Ager*[AHA/+] and *Ager*[AHA/AHA] mice. Our findings suggest that RAGE-AHA mutation led to a haploinsufficient bone phenotype, and that gene dosage of RAGE also plays a role in determining the bone phenotype, especially in female mice.

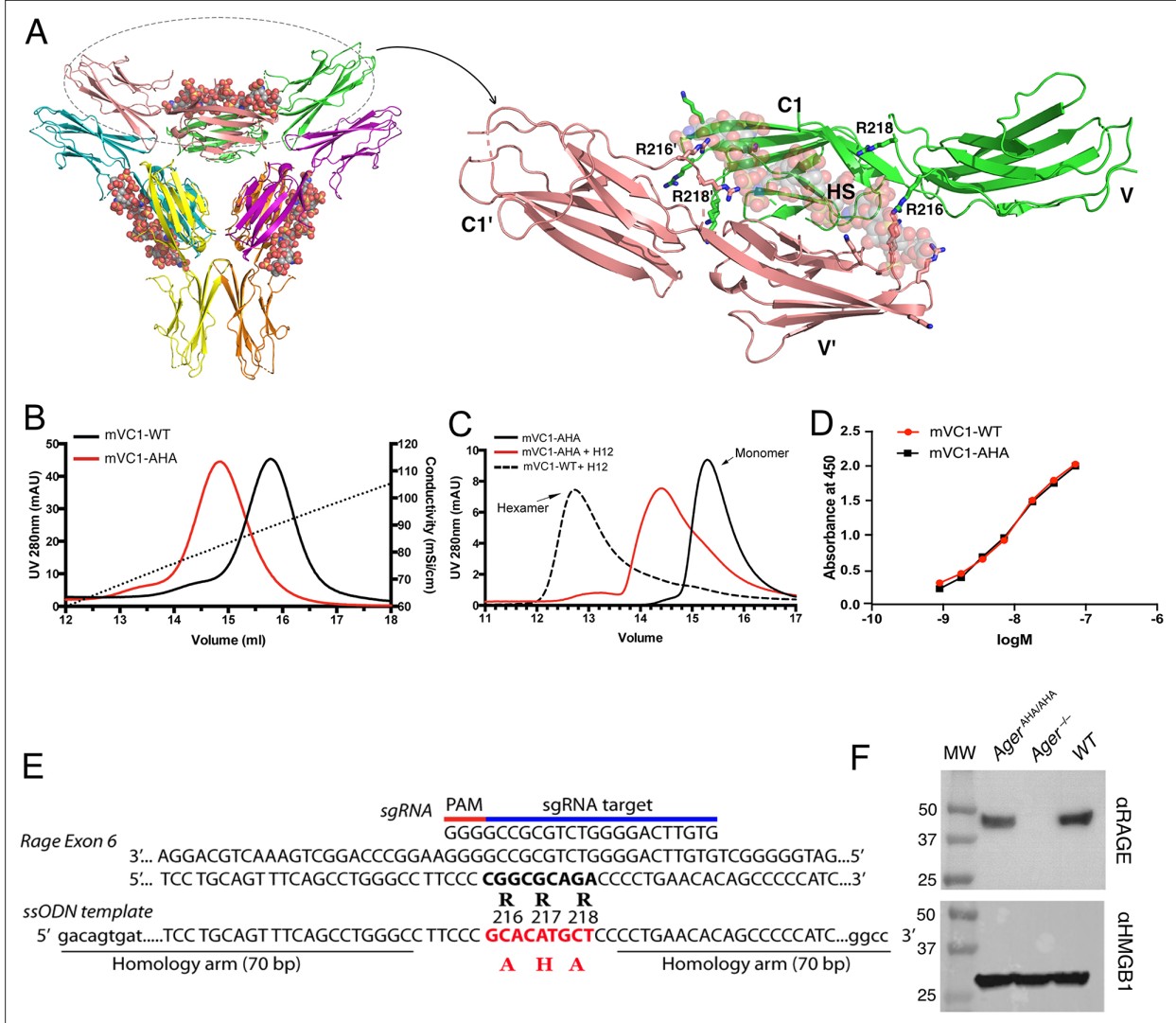

**Figure 1.** Characterization of HS binding-deficient RAGE mutant (R$^{216}$R$^{217}$R$^{218}$ to A$^{216}$H$^{217}$A$^{218}$) and generation of *Ager*$^{AHA/AHA}$ knock-in mice. (**A**) Cartoon diagram of HS-induced hexamer of RAGE V-C1 domains (PDB 4IMB). The hexamer is organized as trimer of dimers, with each dimer stabilized by one molecule of HS oligosaccharide. Oligosaccharides are shown in space-filling models. One dimer (salmon and green) is enlarged to show HS binding residues (in stick representation). R216 and R218 from both monomers (V-C1 and V'-C1') are marked. (**B**) Binding of wild-type murine RAGE V-C1 domain (mVC1-WT) and RAGE-AHA mutant (mVC1-AHA) to heparin Sepharose column. RAGE-AHA mutant had reduced HS-binding capacity. (**C**) mVC1-WT or mVC1-AHA were incubated with HS dodecasaccharide (H12) and the mixtures were resolved on a Superdex 200 (10/300 mm) gel filtration column. H12 was unable to induce RAGE-AHA to form a stable RAGE hexamer. (**D**) Binding affinity of mVC1-WT and mVC1-AHA to immobilized HMGB1 was determined by enzyme-linked immunosorbent assay (ELISA). RAGE-AHA displayed WT-like binding affinity to ligand. (**E**) Targeting strategy for generating *Ager*$^{AHA/AHA}$ knock-in mice. Sequences of the targeting single guide RNA (sgRNA), the mutation sites in *Ager* exon 6, and the repairing template single-stranded donor oligonucleotides (ssODN) are shown. (**F**) Western blotting analysis of RAGE expression in lung lysate. Top panel, lung lysates from WT, *Ager*$^{AHA/AHA}$, and *Ager*$^{-/-}$ mice were blotted with a rat anti-RAGE mAb (R&D system), which showed RAGE was expressed at normal level in *Ager*$^{AHA/AHA}$ mice. Bottom panel, as a protein loading control, the blot was reprobed with anti-HMGB1, which is universally expressed by all cells.

The online version of this article includes the following source data for figure 1:

**Source data 1.** Binding affinity of mVC1-WT and mVC1-AHA to immobilized HMGB1 was determined by enzyme-linked immunosorbent assay (ELISA).

**Source data 2.** Western blotting analysis of RAGE expression in lung lysate.

## Osteoclastogenesis is impaired in *Ager*$^{AHA/AHA}$ mice

Tartrate-resistant acid phosphatase (TRAP) staining of tibia sections showed the number of osteoclasts in *Ager*$^{AHA/AHA}$ mice was reduced to a similar level as in *Ager*$^{-/-}$ mice (*Figure 3A–B*), which indicates a defect of osteoclast differentiation in *Ager*$^{AHA/AHA}$ mice. To confirm this defect, we performed an osteoblast/bone marrow macrophage (BMM) co-culture osteoclastogenesis assay in vitro. In this model, WT

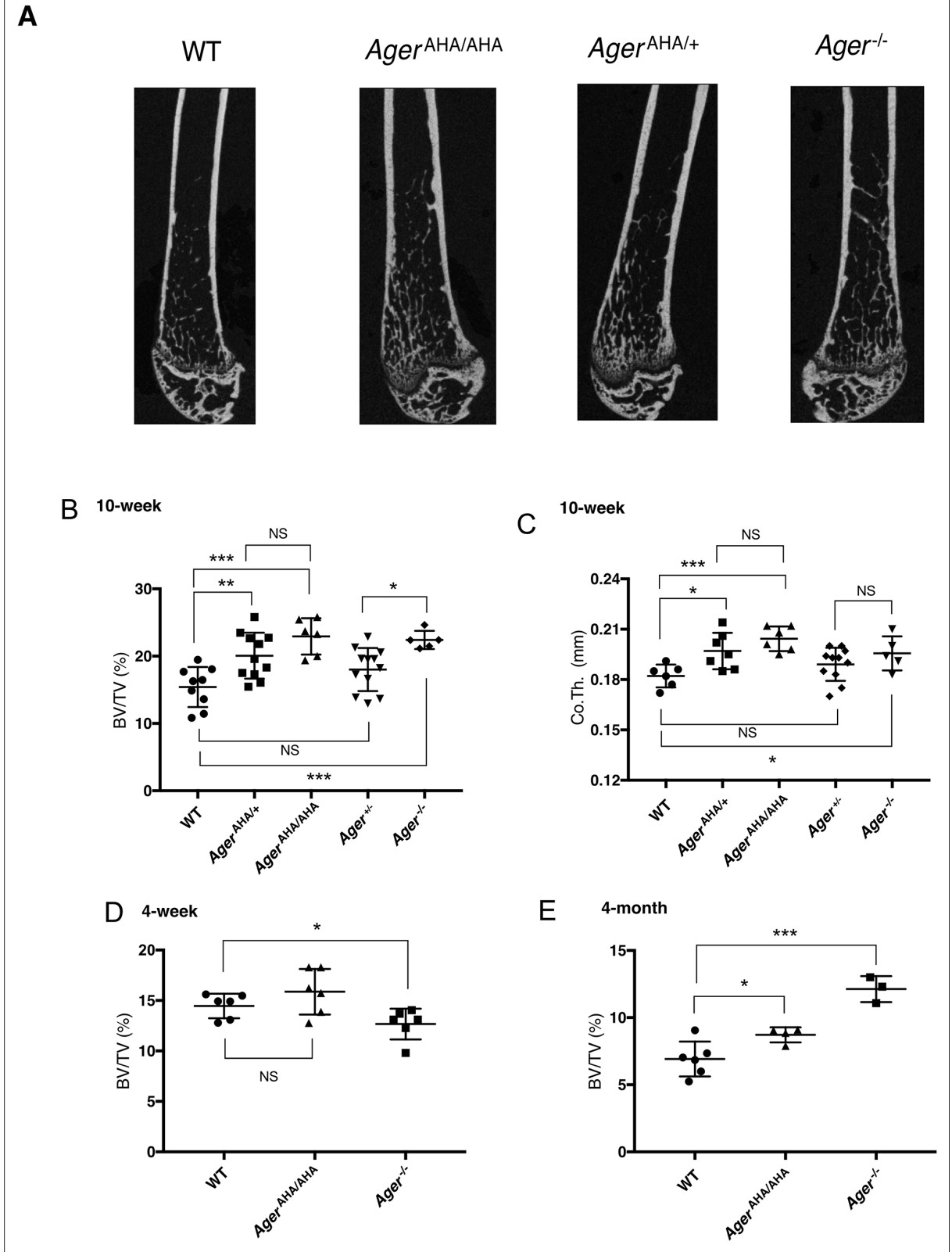

**Figure 2.** $Ager^{AHA/AHA}$ and $Ager^{AHA/+}$ mice develop osteopetrotic phenotype. (**A**) Representative µCT images of the femurs from 10-week-old male WT, $Ager^{AHA/AHA}$, $Ager^{AHA/+}$, and $Ager^{-/-}$ mice. (**B**) Trabecular bone volume/tissue volume ratio (BV/TV), n=5–12, and (**C**) cortical bone thickness of femurs from 10-week-old male WT, $Ager^{AHA/AHA}$, $Ager^{AHA/+}$, $Ager^{+/-}$, and $Ager^{-/-}$ mice, n=5–12. (**D**) Trabecular BV/TV of 4-week-old male WT, $Ager^{AHA/AHA}$, and $Ager^{-/-}$ mice, n=6. (**E**) Trabecular BV/TV of 4-month-old male WT, $Ager^{AHA/AHA}$, and $Ager^{-/-}$ mice, n=3–6. Error bars represent SD. *, **, and *** represent p<0.05,

*Figure 2 continued on next page*

*Figure 2 continued*

0.01, and 0.001, respectively.

The online version of this article includes the following source data and figure supplement(s) for figure 2:

**Source data 1.** *Ager*<sup>AHA/AHA</sup> and *Ager*<sup>AHA/+</sup> mice develop osteopetrotic phenotype.

**Figure supplement 1.** Trabecular bone morphometric analysis of 10-week-old male WT, *Ager*<sup>AHA/AHA</sup>, *Ager*<sup>AHA/+</sup>, *Ager*<sup>−/−</sup>, and Ager<sup>+/−</sup> mice.

**Figure supplement 1—source data 1.** Trabecular bone morphometric analysis of 10-week-old male WT, *Ager*<sup>AHA/AHA</sup>, *Ager*<sup>AHA/+</sup>, *Ager*<sup>−/−</sup>, and Ager<sup>+/−</sup> mice.

**Figure supplement 2.** Female *Ager*<sup>AHA/AHA</sup> and *Ager*<sup>AHA/+</sup> mice develop osteopetrotic phenotype.

**Figure supplement 2—source data 1.** Female *Ager*<sup>AHA/AHA</sup> and *Ager*<sup>AHA/+</sup> mice develop osteopetrotic phenotype.

primary calvarial osteoblasts were co-cultured with BMMs from WT, *Ager*<sup>AHA/AHA</sup>, and *Ager*<sup>−/−</sup> mice to induce osteoclastogenesis from BMMs. Compared to the large and multinucleated TRAP-positive cells formed in cultures of WT BMMs, *Ager*<sup>AHA/AHA</sup>, and *Ager*<sup>−/−</sup> formed much smaller TRAP-positive cells (*Figure 3C*). To better quantify the differences in osteoclast numbers among the three genotypes, we sub-grouped the osteoclasts based on the number of nuclei number they contain (*Figure 3D*). Interestingly, while the number smaller osteoclasts (3–10 nuclei) were not significantly reduced in *Ager*<sup>AHA/AHA</sup> and *Ager*<sup>−/−</sup>, the number of larger osteoclasts (>11 nuclei) were substantially reduced (*Figure 3D*). Collectively, these results demonstrate that impaired HS-RAGE interaction leads to abnormal maturation of osteoclasts, which suggests a key role for RAGE-HS interaction in regulating osteoclastic differentiation and function. In addition, the mutagenesis in osteoblast lineage cells did not seem to have an impact on bone formation as examined by fluorochrome double labeling (*Figure 3—figure supplement 1*).

## *Ager*<sup>AHA/AHA</sup> mice were protected from liver injury after APAP overdose

We and others have previously reported that RAGE was responsible for neutrophil-mediated secondary injury (in an HMGB1-dependent manner) following acetaminophen (APAP)-induced liver necrosis (*Arnold et al., 2020*; *Huebener et al., 2015*). When subjected to a sublethal dose of APAP, *Ager*<sup>−/−</sup> mice displayed a reduction of neutrophil infiltration and liver damage compared with WT mice. We thus similarly used this APAP overdose model to examine the role of HS-RAGE interaction in mediating neutrophilic liver injury. As expected, we found that like *Ager*<sup>−/−</sup> mice, *Ager*<sup>AHA/AHA</sup> mice were also protected from APAP-induced liver injury. The plasma level of alanine aminotransferase (ALT), a biomarker of liver damage, was reduced by nearly 45% in *Ager*<sup>AHA/AHA</sup> mice (*Figure 4A*). Similarly, neutrophil infiltration into the liver was also significantly reduced in *Ager*<sup>AHA/AHA</sup> mice (*Figure 4B*). Histologic examination also confirmed that livers from *Ager*<sup>AHA/AHA</sup> mice had significantly reduced necrotic area compared to the injured WT livers (*Figure 4C–D*). Notably, these results were very similar to what we previously observed in *Ager*<sup>−/−</sup> mice (*Arnold et al., 2020*). Our data strongly suggest that in this drug-induced liver injury model, HS-RAGE interaction plays an essential role in mediating HMGB1-dependent neutrophil infiltration.

## RNA-seq transcriptome analysis of neutrophils from *Ager*<sup>AHA/AHA</sup> mice and *Ager*<sup>−/−</sup> mice

To understand whether ablation of RAGE signaling has an impact on the general mobility of neutrophils, we performed an air pouch assay to examine LPS-induced neutrophil infiltration. Interestingly, in this assay, neutrophil infiltration was indistinguishable between WT, *Ager*<sup>AHA/AHA</sup>, and *Ager*<sup>−/−</sup> mice (*Figure 5A*). This result confirmed that *Ager*<sup>AHA/AHA</sup> and *Ager*<sup>−/−</sup> neutrophils have normal mobility when a different chemoattractant (other than HMGB1) was responsible for the infiltration. During this assay, we unexpectedly discovered that several common surface markers were dramatically upregulated on *Ager*<sup>−/−</sup> neutrophils with 100% penetrance. Compared to WT neutrophils, the expression level of Ly6C and CD45 on *Ager*<sup>−/−</sup> neutrophils were 10- to 15-fold higher (*Figure 5B*). Surprisingly, the expression levels of both markers were completely normal on *Ager*<sup>AHA/AHA</sup> neutrophils. The abnormal expression of Ly6C and CD45 was not restricted to neutrophils as we also found they are overexpressed in T and B cells from *Ager*<sup>−/−</sup> mice (*Figure 5—figure supplement 1*). The fact that two out of eight common leukocyte markers were highly elevated in *Ager*<sup>−/−</sup> (the other six markers were normal, including CD11b shown in *Figure 5B*), but were normal in *Ager*<sup>AHA/AHA</sup> neutrophils strongly suggests that complete

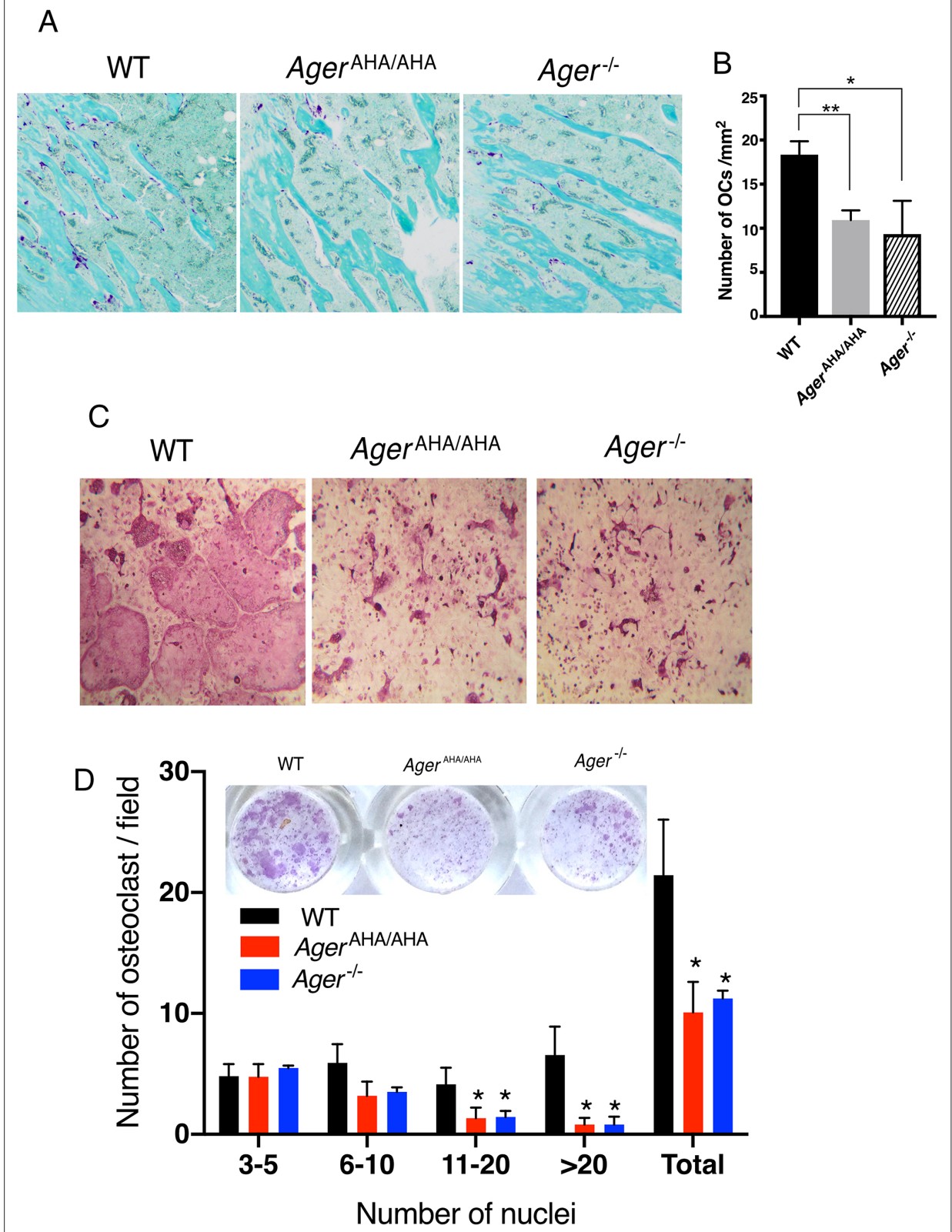

**Figure 3.** $Ager^{AHA/AHA}$ mice display impaired osteoclastogenesis. (**A**) TRAP staining of paraffin sections of WT, $Ager^{AHA/AHA}$, and $Ager^{-/-}$ tibias. Mature osteoclasts were stained purple, and bone was counterstained with Fast Green. (**B**) Quantification of osteoclasts (OCs) in tibia sections. n=3 mice. (**C**) BMM isolated from WT, $Ager^{AHA/AHA}$, or $Ager^{-/-}$ mice was co-cultured with WT osteoblasts. Mature osteoclasts are visualized by TRAP staining. (**D**) Quantification of the number of osteoclasts per 100× field. Osteoclasts were sub-grouped into four categories based on the number of nuclei they

*Figure 3 continued on next page*

*Figure 3 continued*

contain. n=3 wells. * and ** represent p<0.05 and 0.01, respectively. Data are representative of at least three separate assays. TRAP, tartrate-resistant acid phosphatase; WT, wild-type.

The online version of this article includes the following source data and figure supplement(s) for figure 3:

**Source data 1.** *Ager*[AHA/AHA] mice display impaired osteoclastogenesis.

**Figure supplement 1.** Bone formation was unaltered in *Ager*[AHA/AHA] and *Ager*[−/−] mice.

**Figure supplement 1—source data 1.** Mineral apposition rate (MAR) calculated from the double-labeling analysis in 8-week-old WT, *Ager*[AHA/AHA], and *Ager*[−/−] tibia.

removal of RAGE protein has a much broader global impact compared to mere point mutations of RAGE. To grasp the full scale of the alteration in gene expression caused by knocking out *Rage,* we performed an RNA-seq analysis of mature neutrophils from WT, *Ager*[AHA/AHA], and *Ager*[−/−] mice.

Principle component analysis of the RNA-seq data found clear separation of transcriptomes of *Ager*[AHA/AHA], *Ager*[−/−], and WT mice, with all three biological replicates of each genotype closely clustering together (*Figure 5—figure supplement 2*). As expected, the overall number of differentially expressed genes (DEGs) in *Ager*[−/−] neutrophils were almost 2.5 times higher than in *Ager*[AHA/AHA]

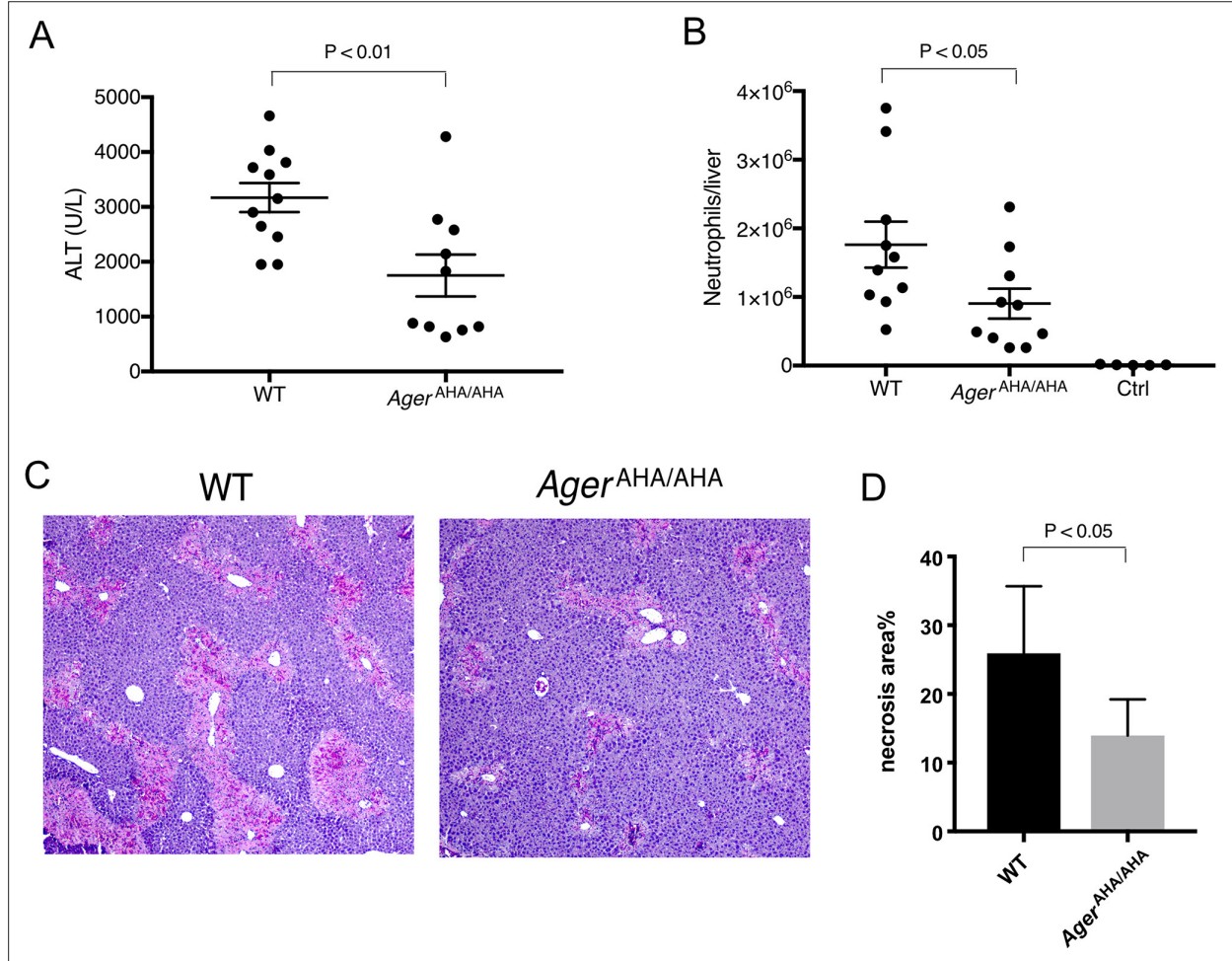

**Figure 4.** *Ager*[AHA/AHA] mice were protected from liver injury after APAP overdose. (**A**) WT and *Ager*[AHA/AHA] mice were treated with 300 mg/kg APAP to induce liver injury and plasma ALT concentrations were measured 24 hr post injury, n=10–11. (**B**) Neutrophils recruitment into the liver after APAP-induced liver injury, n=5–10. (**C**) Hematoxylin and eosin (H&E) staining of paraffin-embedded liver tissues to show the extent of liver necrosis. (**D**) The necrotic area of H&E stained liver tissues was quantified with ImageJ, n=5.

The online version of this article includes the following source data for figure 4:

**Source data 1.** *Ager*[AHA/AHA] mice were protected from liver injury after APAP overdose.

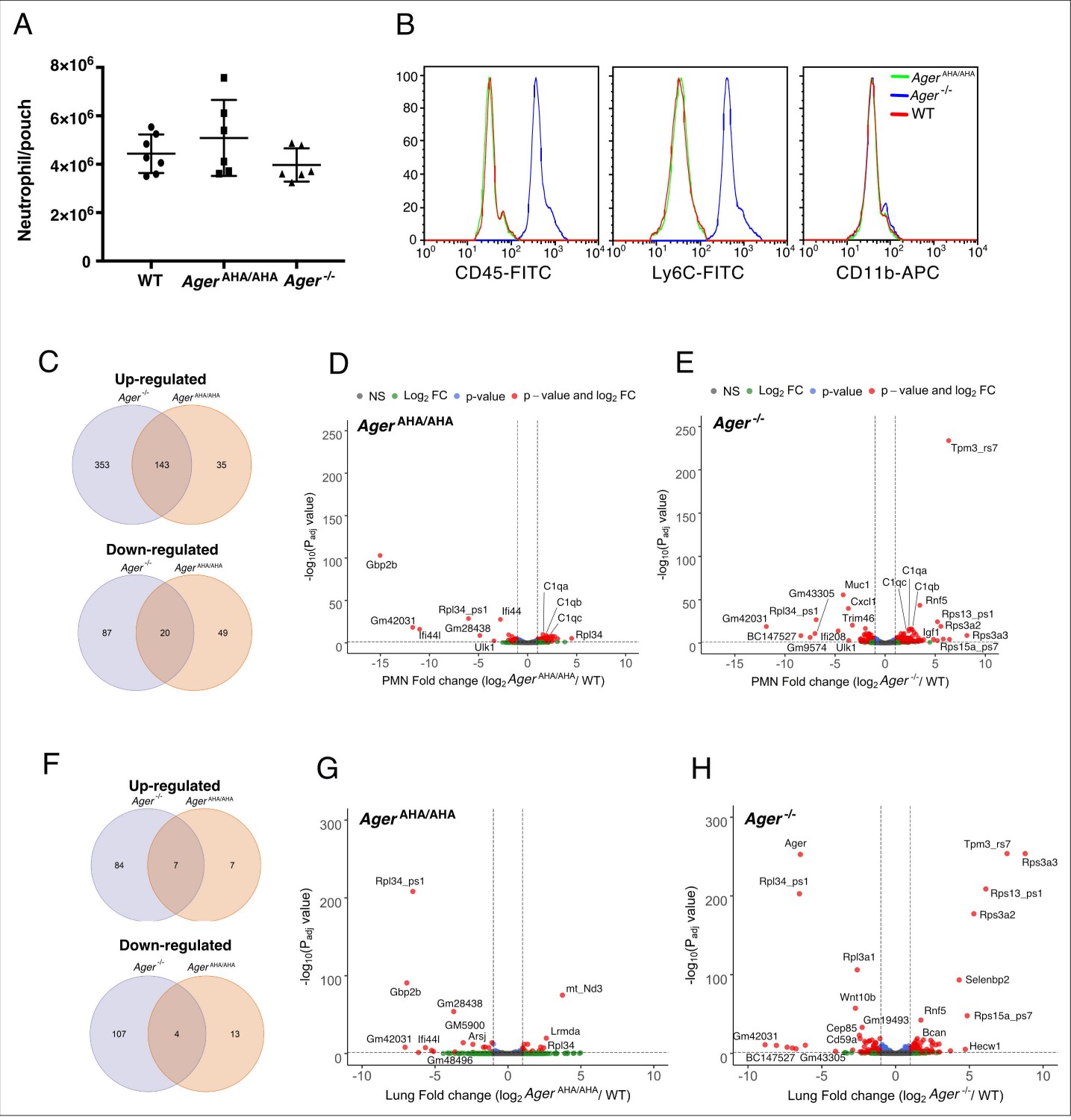

**Figure 5.** RNA-seq transcriptome analysis of *Ager*[AHA/AHA] mice and *Ager*[−/−] mice. (**A**) LPS-induced neutrophil infiltration into air pouch in WT, *Ager*[AHA/AHA], and *Ager*[−/−] mice, n=6–7. Neutrophils were quantified 4 hr after LPS injection. (**B**) Surface marker (CD45, Ly6C, and CD11b) analysis of air pouch neutrophils by FACS. Cells from six mice of each genotype were analyzed with identical result. (**C–E**) RNA-seq transcriptome analysis of *Ager*[AHA/AHA] and *Ager*[−/−] mice PMN. (**C**) Plot of up- and downregulated genes in *Ager*[AHA/AHA] and *Ager*[−/−] PMN. DEGs between *Ager*[AHA/AHA] and WT PMNs were determined based on Benjamin–Hochberg multicomparison correction p values. 178 genes were found to be significantly upregulated ($p_{adj}$ value≤0.05, $\log_2$ fold change≥1) while 69 genes were found to be significantly downregulated ($p_{adj}$ value≤0.05, $\log_2$ fold change≤–1). DEGs between *Ager*[−/−] and WT PMNs were determined similarly. 476 genes were found to be significantly upregulated ($p_{adj}$ value≤0.05, $\log_2$ fold change≥1) while 107 genes were found to be significantly downregulated ($p_{adj}$ value≤0.05, $\log_2$ fold change≤–1). (**D**) Volcano plot of DEGs between *Ager*[AHA/AHA] PMN and WT PMNs. Genes with

*Figure 5 continued on next page*

*Figure 5 continued*

most significant changes were marked in the plot. (**E**) Volcano plot of DEGs between *Ager*^−/− PMN and WT PMNs. (**F–H**) RNA-seq transcriptome analysis of whole lungs from *Ager*^AHA/AHA and *Ager*^−/− mice. (**F**) Plot of up- and downregulated genes in *Ager*^AHA/AHA and *Ager*^−/− lungs. 14 genes were found to be significantly upregulated while 17 genes were found to be significantly downregulated in *Ager*^AHA/AHA lungs. In *Ager*^−/− lungs, 91 genes were found to be significantly upregulated while 111 genes were found to be significantly downregulated. (**G**) Volcano plot of DEGs between *Ager*^AHA/AHA and WT lungs. Genes with most significant changes were marked in the plot. (**H**) Volcano plot of DEGs between *Ager*^−/− and WT lungs. DEG, differentially expressed gene; PMN, polymorphonuclear; WT, wild-type.

The online version of this article includes the following source data and figure supplement(s) for figure 5:

**Source data 1.** LPS-induced neutrophil infiltration into air pouch in WT, *Ager*^AHA/AHA , and *Ager*^−/− mice.

**Figure supplement 1.** T cells and B cells from *Ager*^−/− mice display greatly elevated surface expression of Ly6C and CD45.

**Figure supplement 2.** Principal component analysis of RNAseq data.

**Figure supplement 3.** Expression levels of selected genes as determined by RNA-seq.

**Figure supplement 3—source data 1.** Expression levels of selected genes as determined by RNA-seq.

**Figure supplement 4.** Expression levels of highly expressed AGER ligands as determined by RNA-seq.

**Figure supplement 4—source data 1.** Expression levels of highly expressed AGER ligands as determined by RNA-seq.

neutrophils (603 vs. 247). Significant overlap was found in upregulated genes, with 80% of upregulated genes in *Ager*^AHA/AHA (143/178) also being upregulated in *Ager*^−/− neutrophils (*Figure 5C* and *Supplementary file 1*). In contrast, the overlap was rather limited in downregulated genes, with merely 29% (20/69) of downregulated genes in *Ager*^AHA/AHA also being downregulated in *Ager*^−/−. Gene ontology (GO) analysis of upregulated genes found that genes related to regulation of leukocyte activation (43 genes), differentiation (27 genes), proliferation (29 genes), adhesion (34 genes), and regulation of cytokine production (30 genes) were highly enriched in *Ager*^−/− neutrophils (*Supplementary file 2*). The same biological processes were also enriched in *Ager*^AHA/AHA neutrophils, except that the number of genes involved were roughly half of the number of genes found in *Ager*^−/− neutrophils (*Supplementary file 3*). These findings suggest that RAGE signaling plays a profound role in leukocyte biology. GO analysis of downregulated genes did not find any biological processes that are significantly differentially regulated in either *Ager*^−/− and *Ager*^AHA/AHA neutrophils. Curiously, we did identify several highly downregulated genes in *Ager*^AHA/AHA neutrophils that were not found in *Ager*^−/− neutrophils, including Gbp2b, Ifi44, and Ifi44l (*Figure 5D* and *Figure 5—figure supplement 3A-C*), all of which have been shown to be interferon γ-induced genes and play important roles in defense against infection (*Santos and Broz, 2018*; *Kim et al., 2011*; *Busse et al., 2020*). Many top DEGs in *Ager*^−/− neutrophils were not found in *Ager*^AHA/AHA neutrophils (*Figure 5E*), which include highly upregulated genes: Rps3a2, Rps3a3, Rps13_ps1, Rps15a_ps7, Tpm3_rs7, and Rnf5; and highly downregulated genes: Cxcl1, Muc1, Trim46, Ifi208, and BC147527. In addition, we also examined expression levels of seven most highly expressed RAGE ligands and found none of those were altered in *Ager*^AHA/AHA neutrophils (*Figure 5—figure supplement 4*). In *Ager*^−/− neutrophils, while we found expression levels of most RAGE ligands were unaltered, the expression levels of S100a8 and S100a11 were reduced by ~40%. To our surprise, Ly6C and CD45 were not among the DEGs in *Ager*^−/− neutrophils, which suggest that either the posttranscriptional regulation or the secretion of Ly6C and CD45 were altered in *Ager*^−/− neutrophils. If that were the case, we would expect that the global protein expression profile of *Ager*^−/− neutrophils would perhaps be even more dramatically altered than what is apparent on RNA sequencing.

## RNA-seq transcriptome analysis of lung tissues from *Ager*^AHA/AHA mice and *Ager*^−/− mice

To gain further insight into the impact of impaired RAGE signaling on gene transcription, we performed RNA-seq analysis of lung tissue, an organ in which RAGE is abundantly expressed. Surprisingly, the number of DEGs was much less in lungs compared to neutrophils in both strains. We found 202 DEGs in *Ager*^−/− lungs and merely 31 DEGs in *Ager*^AHA/AHA lungs (*Figure 5F* and *Supplementary file 4*). The difference between the two genotypes was however even more dramatic in lungs (7-fold) than we observed in neutrophils (2.5-fold). Similar to what we found in neutrophils, in *Ager*^AHA/AHA lung significant overlap was found in upregulated genes, with 50% of upregulated genes in *Ager*^AHA/AHA (7/14) were also upregulated in *Ager*^−/− (*Figure 5F* and *Supplementary file 4*). In contrast, the overlap was

very limited in downregulated genes, with merely 23.5% (4/17) of downregulated genes in $Ager^{AHA/AHA}$ were similarly downregulated in $Ager^{-/-}$. Comparing the DEGs between lungs and neutrophils, we found that interestingly, many top DEGs appeared in both tissues in a genotype-specific manner. In $Ager^{AHA/AHA}$, Gbp2b, Ifi44, and Gm28438 were highly downregulated in both lungs and neutrophils (*Figure 5D and G*, *Figure 5—figure supplement 3*). In $Ager^{-/-}$, Rps3a2, Rps3a3, Rps13_ps1, Rps15a_ps7, Tpm3_rs7, Rf5, and BC147527 were differently regulated in both lungs and neutrophils to similar extent (*Figure 5E and H*). Furthermore, two genes, Rpl34_ps1 and Gm42031, were found to be highly downregulated in both genotypes and in both tissues. As expected, in $Ager^{-/-}$ lung, Ager (the gene that expresses RAGE) is one of the top downregulated genes (*Figure 5H*). Also consistent with our Western blot analysis (*Figure 1F*), in $Ager^{AHA/AHA}$ lung, Ager was expressed at a similar level as in WT lung on RNA sequencing (*Figure 5—figure supplement 3F*). GO analysis revealed genes related to regulation of cell killing (11 genes), canonical Wnt signaling (7 genes), and epithelial cell proliferation (11 genes), among others, was highly enriched in $Ager^{-/-}$ lung (*Supplementary file 5*). In contrast, none of these biological processes were enriched in $Ager^{AHA/AHA}$ lung due to limited number of DEGs.

## Development of a rabbit mAb that binds specifically to the HS-binding site of RAGE

As $Ager^{AHA/AHA}$ mice phenocopy $Ager^{-/-}$ mice in both osteoclastogenesis and response to APAP-induced liver injury, inhibition of HS-RAGE interaction could be an effective means to block RAGE signaling. To this end, we developed a rabbit monoclonal antibody (mAb B2) to specifically target HS-RAGE interaction. The epitope of B2 includes the HS-binding residues R216 and R218. Compared to the apparent binding affinity between B2 and WT RAGE (0.1 nM), the affinity between B2 and R216A-R218A mutant was dramatically reduced to 4 nM (*Figure 6A*). Using a heparin Sepharose column, we confirmed that B2 was indeed able to significantly inhibit the binding of RAGE to heparin (*Figure 6B*). Importantly, while a polyclonal anti-RAGE antibody significantly inhibited the binding between RAGE and its ligands HMGB1 and S100B, B2 did not interfere with the binding between RAGE and its ligands (*Figure 6C–D*). The specificity of B2 was determined by immunostaining of lung sections from WT and $Ager^{-/-}$ mice. RAGE-specific staining was observed in WT lung section while no staining was shown on $Ager^{-/-}$ (*Figure 6E*). Western blot analysis of whole lung lysate from WT, $Ager^{AHA/AHA}$, and $Ager^{-/-}$ mice further showed that B2 was highly specific for RAGE but failed to detect the RAGE-AHA mutant, which again confirmed the epitope of B2 includes R216 and R218 (*Figure 6F*).

Using a human breast cancer MDA-MB-453 cell line, which was known to overexpress RAGE (*Nasser et al., 2015*), we examined B2 binding to cell surface RAGE by FACS. As expected, B2 stains 453 cells extremely well, displayed a two-log shift compared to IgG control (*Figure 6G*). Interestingly, when 453 cells were pretreated with heparin lyase III (HL-III) to remove cell surface HS, binding of B2 IgG was dramatically reduced by 20-fold (*Figure 6G*). Because removal of HS would dissociate cell surface oligomeric RAGE back to monomers, we suspect that B2 preferably binds to dimeric RAGE in a bivalent fashion. Indeed, when Fab fragment of B2 was tested on 453 cells, we found that its cell surface binding was much weaker (due to monovalent interaction), and the binding showed minimal sensitivity to HL-III treatment (*Figure 6H*).

## B2 inhibits RAGE-dependent biological processes in cell and animal models

We first tested the inhibitory activity of B2 on RAGE-dependent osteoclastogenesis. BMMs were co-cultured with primary calvarial osteoblasts in the presence of 20 µg/ml of B2 or isotype-matched control antibody. As expected, cells treated with B2 formed fewer and smaller TRAP-positive cells, especially for cells with more than 10 nuclei (*Figure 7A–B*). Remarkably, the osteoclastogenic phenotype of WT BMM treated with B2 was essentially identical to that of $Ager^{AHA/AHA}$ BMMs (*Figure 3C–D*). In monoculture osteoclastogenesis assay, where BMMs were induced with m-CSF and RANKL, B2 was equally effective in inhibiting osteoclast formation (*Figure 7C*). The hepatoprotective effects of B2 were also examined in an APAP-induced acute liver failure model. In this experiment, mice were pretreated with B2 at 10 mg/kg before injection of sublethal dose of APAP. Compared to control IgG treated mice, B2 was able to reduce plasma ALT concentration by 40% and neutrophil infiltration by 50% (*Figure 7D-E*). Again, the level of protection provided by B2 was highly similar to what

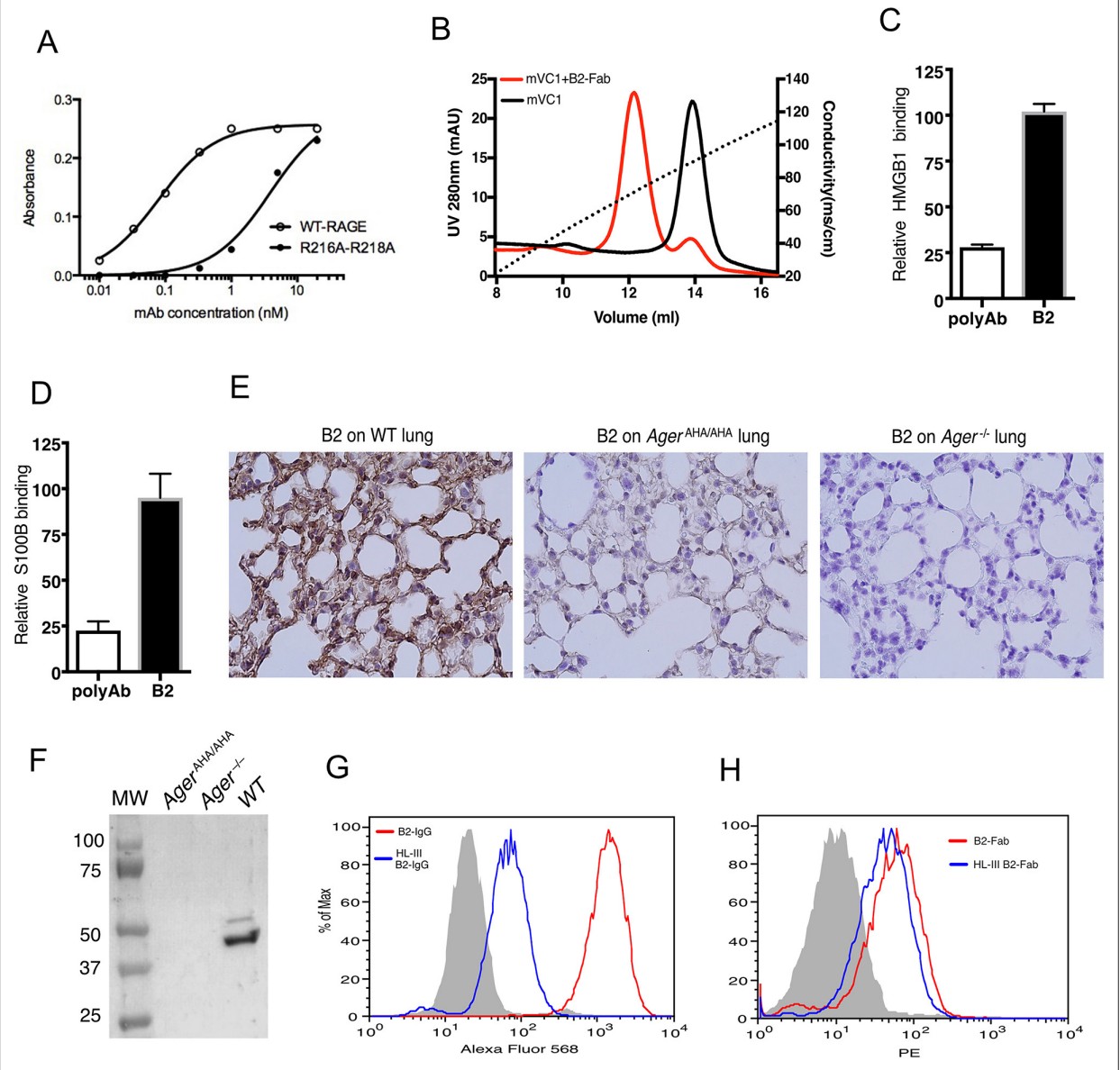

**Figure 6.** Development of a rabbit mAb that targets specifically to the HS-binding site of RAGE. (**A**) Binding of rabbit mAb B2 to immobilized wild-type (WT) sRAGE or R216A-R218A mutant was determined by ELISA. The dramatically reduced binding affinity of B2 to R216A-R218A mutant ($K_d$=4 nM) compared to its affinity to WT sRAGE ($K_d$=0.1 nM) indicates that R216 and R218 are part of the epitope for B2. (**B**) B2 directly inhibits binding of RAGE to heparin. WT mouse RAGE VC1 domain (mVC1) were either directly loaded onto heparin Sepharose column, or loaded after 30 min incubation with Fab fragment of B2 at 1:1 molar ratio. B2-Fab bound mVC1 displayed greatly reduced binding to heparin column. (**C**) Binding of biotinylated sRAGE to immobilized HMGB1 was measured in the presence of anti-RAGE rabbit polyclonal Ab or B2 at 5 µg/ml. Binding of sRAGE to HMGB1 in the absence of antibodies was set to 100%. (**D**) Binding of biotinylated sRAGE to immobilized S100B was measured in the presence of anti-RAGE rabbit polyclonal Ab or B2 at 5 µg/ml. (**E**) Determine the specificity of B2 by immunostaining of lung sections from WT, *Ager*^AHA/AHA^, and *Ager*^−/−^ mice with 1 µg/ml B2. (**F**) Western blot analysis of the specificity of B2 using lung lysate from WT, *Ager*^AHA/AHA^, and *Ager*^−/−^ mice. Please note B2 failed to detect RAGE-AHA mutant, again confirming the epitope of B2. The loading ctrl is the same as shown in ***Figure 1F***. (**G**) Binding of B2 IgG to untreated MDA-453 cells or cells pretreated with heparin lyase III (HL-III). Whole IgG form of B2 (bivalent) binds to cell surface RAGE in an HS-dependent manner. (**H**) Binding of B2 Fab fragment to untreated MDA-453 cells or cells pretreated with HL-III. Binding of the Fab form of B2 (monovalent) to RAGE was not sensitive to HL-III treatment. The shaded histograms in (**G–H**) are from cells stained only with control antibodies.

The online version of this article includes the following source data for figure 6:

**Source data 1.** Development of a rabbit mAb that targets specifically to the HS-binding site of RAGE.

**Source data 2.** Western blot analysis of the specificity of B2 using lung lysate from WT, *Ager*^AHA/AHA^, and *Ager*^−/−^ mice.

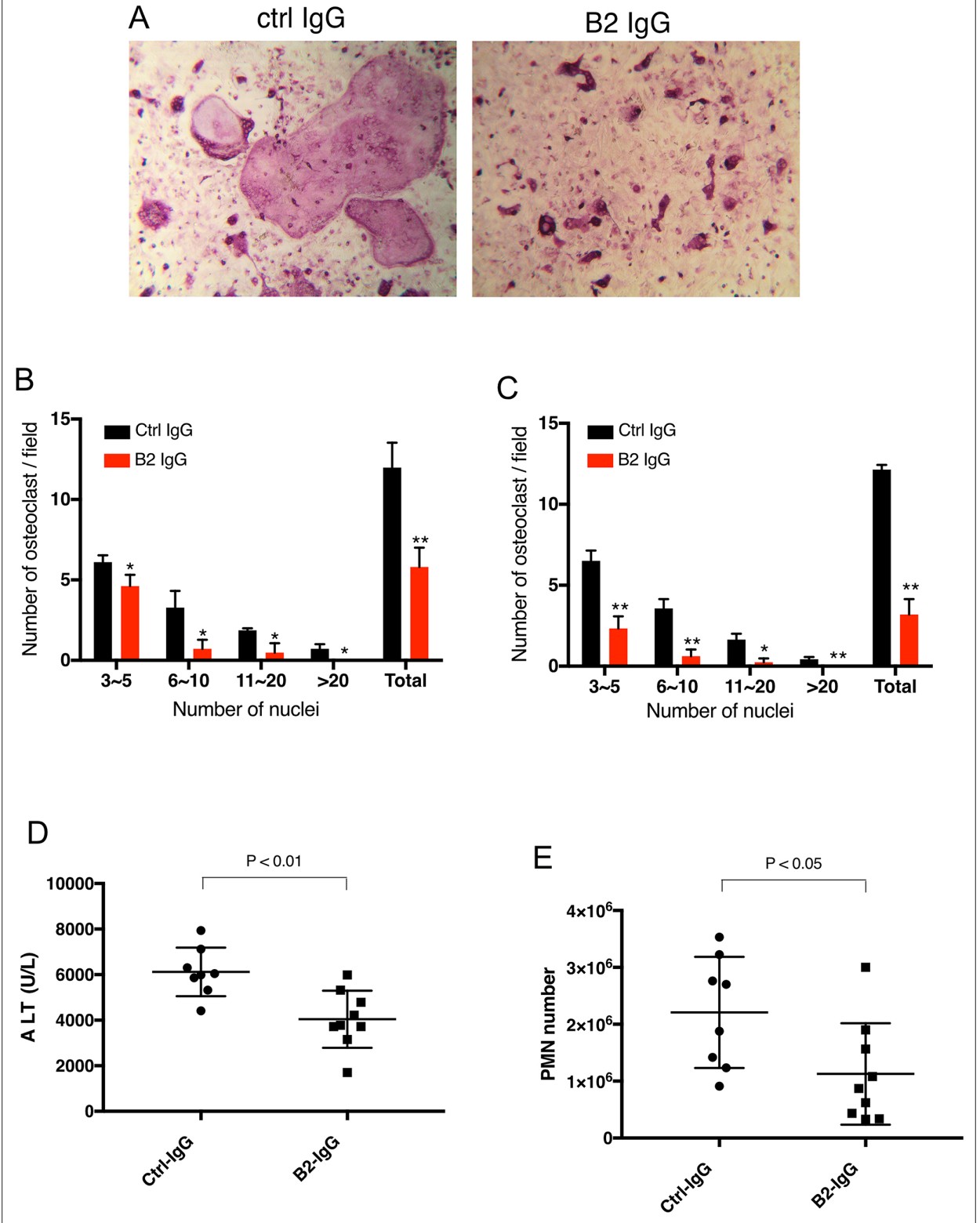

**Figure 7.** B2 inhibits RAGE-dependent biological processes in cell and animal models. (**A**) B2 inhibits osteoclastogenesis in vitro. Representative images of TRAP staining of WT bone marrow cells co-cultured with WT osteoblasts in the presence of 20 µg/ml B2 or control IgG. Mature osteoclasts were stained pink. (**B**) Osteoclasts were sub-grouped into four categories based on the number of nuclei and quantified, n=3 wells. (**C**) B2 also inhibits osteoclastogenesis in monoculture assay, where differentiation of bone marrow macrophages was induced by exogenous RANKL and m-CSF, n=3 wells. (**D, E**) B2 was protective as a pretreatment in APAP-induced liver injury. WT mice were pretreated with 10 mg/kg of B2-IgG 12 h before APAP overdose

*Figure 7 continued on next page*

*Figure 7 continued*

(300 mg/kg). Plasma ALT concentrations (**D**) and neutrophils recruitment into to liver, n=8–9 (**E**) were quantified 24 hr post liver injury, n=8–9. TRAP, tartrate-resistant acid phosphatase; WT, wild-type.

The online version of this article includes the following source data for figure 7:

**Source data 1.** B2 inhibits RAGE-dependent biological processes in cell and animal models.

was observed in *Ager*$^{AHA/AHA}$ mice (*Figure 4A–B*). Collectively, these results validated our strategy of targeting the HS-binding site of RAGE as a means to inhibit RAGE signaling.

## Discussion

While our previous work has shown that HS is a potent inducer of RAGE oligomerization and that HS-RAGE interaction is essential for RAGE signaling in endothelial cells (*Xu et al., 2013*), the physiological significance of HS-induced RAGE oligomerization remains unexplored due to a lack of genetic models. To address this gap, we generated a *Ager*$^{AHA/AHA}$ knock-in mice to specifically disrupt HS-RAGE interactions. As the HS-binding site and the ligand-binding site of RAGE are spatially separated (*Xu et al., 2013*), we were able to create a RAGE variant (R216A-R217H-R218A) with normal ligand binding, but impaired HS-binding capacity (*Figure 1*). This novel mouse model allowed us to dissect the specific contribution of HS-RAGE interaction to RAGE signaling in both physiological and pathological conditions for the first time.

*Zhou et al., 2006* first reported that *Ager*$^{-/-}$ mice display enhanced bone mineral density due to impaired osteoclastogenesis. Examination of the bone morphometric indexes of *Ager*$^{AHA/AHA}$ mice found that they also display an osteopetrotic bone phenotype in both trabecular and cortical bones, with a severity that was equivalent to *Ager*$^{-/-}$ mice (*Figure 2B*). Natural history study of the bone phenotype found that for both strains, the bone was largely normal at 4 weeks old (*Figure 2D*), suggesting the osteoclastogenesis during the bone modeling phase is not dependent on RAGE signaling. The osteopetrotic phenotype became full-fledged when mice reach 10 weeks old (for both sexes), and the phenotype remained when mice reach 4 months of age. It is interesting that while the bone phenotypes of *Ager*$^{AHA/AHA}$ and *Ager*$^{-/-}$ mice were indistinguishable at 10 weeks for both males and females, their phenotypes at 4 weeks and 4 months were somewhat different. It is unclear what factors contributed to this, but we suspect that the extensive transcriptomic changes experienced by *Ager*$^{-/-}$ mice (*Figure 5*) might have modified the bone phenotype in an age-dependent manner. The fact that RAGE signaling regulates bone density in mature bones only, and that it regulates both trabecular and cortical bone homeostasis suggests that RAGE would be an attractive therapeutic target for treating diseases involving abnormal bone remodeling.

The fact that the *Ager*$^{AHA/+}$ mice are haploinsufficient suggests that RAGE$^{AHA}$ mutant protein plays a dominant effect on WT RAGE. The dominant negative role of RAGE$^{AHA}$ mutant can only be explained in the light of RAGE oligomerization. In heterozygous *Ager*$^{AHA/+}$ mice, the chance of assembling a functional dimer with intact HS-binding site would be merely 25% (50%×50%). This represents a 75% loss of functional RAGE signaling complex compared to WT mice, which apparently was sufficiently severe to impair normal osteoclastogenesis. In contrast, in *Ager*$^{+/-}$ mice, because all expressed RAGE are WT, they have a 100% of chance to form functional RAGE signaling complex. The fact that male *Ager*$^{+/-}$ mice displayed slightly elevated bone density (although not significant) could be attributed to somewhat reduced overall expression level of RAGE, which can be reasonably expected from heterozygous mice. Interestingly, this gene dosage effect was in full play in 10-week-old female mice, where the *Ager*$^{+/-}$ mice did show significantly increased bone volume (49%), which was identical to the increase observed in *Ager*$^{AHA/+}$ mice (51%). In summary, our analysis has shown for the first time that mice are highly sensitive to the quantity of functional RAGE oligomeric complex, and that female mice appeared to be more sensitive to this regulation than males.

Our genetic manipulation of HS-RAGE interaction has provided clear evidence that disrupting HS-RAGE interaction pharmacologically would be an effective means to block RAGE signaling. To validate this novel concept, we developed a mAb (B2) that specifically targets the HS-binding site of RAGE. The epitope of B2 includes R216 and R218, the exact residues that we mutated in *Ager*$^{AHA/AHA}$ mice. Accordingly, B2 treatment of WT BMMs precisely copied the osteoclastogenesis phenotype of *Ager*$^{AHA/AHA}$ BMMs (*Figures 7A–C , and 3C–D*). Similarly, B2 treatment also provided

significant protection against APAP-induced liver injury, and the extent of protection in liver damage and suppression in neutrophil infiltration was highly similar to what was observed in *Ager*^AHA/AHA^ mice (*Figures 7D–E , and 4*). Combined, these data show that HS-RAGE interaction can be effectively targeted pharmacologically.

From a pharmacological point of view, targeting the HS-binding site of RAGE bears two main advantages over strategies targeting the ligand-binding site of RAGE. First, targeting the HS-binding site of RAGE can in principle block the signaling of all RAGE ligands because HS-dependent RAGE oligomerization is a common mechanism of RAGE signaling irrespective of the ligands involved. This is difficult to achieve for inhibitors that target the ligand-binding site of RAGE because the binding sites for different RAGE ligands are not identical (*Koch et al., 2010*; *Xue et al., 2011*; *Ostendorp et al., 2007*; *Leclerc et al., 2007*). Second, inhibitors that target the HS-binding site of RAGE leave the ligand-binding site of RAGE unoccupied, which would be free to interact with its multiple ligands that are overexpressed in disease states. Such inhibitors may thus turn RAGE into a decoy receptor, which is able to soak up ligands without transducing signal. In principle, this would greatly lower the risk of undesirable side effects by preventing spillover of unbound RAGE ligands to other inflammatory receptors such as TLR4, with which RAGE share many ligands (*Yang et al., 2010*; *Vogl et al., 2007*; *Foell et al., 2013*; *Esposito et al., 2014*).

By comparing transcriptomes of neutrophils and lung tissues from *Ager*^AHA/AHA^ and *Ager*^−/−^ mice, we present here clear evidence that complete deficiency of RAGE had much broader impact on global gene expression compared to point mutations of RAGE. Under the assumption that RAGE signaling is abolished in *Ager*^AHA/AHA^ mice, which was true in both animal models we tested, we predict that most of the additional DEGs found in *Ager*^−/−^ mice were perhaps not associated with RAGE signaling. Presumably, the expression levels of these genes were changed due to certain compensatory mechanisms for the sheer loss of RAGE protein. Because some of the changes in expression level were truly dramatic in both mRNA and protein level, it is highly plausible that some of these changes will manifest as phenotypes in *Ager*^−/−^ mice in certain disease models, but in reality might have little to do with RAGE signaling. While it can be argued that the additional DEGs found in *Ager*^−/−^ mice are due to HS-independent RAGE signaling, such position cannot be well defended for the following reason. If RAGE signaling really consists of HS-independent and HS-dependent RAGE signaling (which is abolished in *Ager*^AHA/AHA^ mice), one would expect that the DEGs in *Ager*^AHA/AHA^ mice are fully encompassed by the DEGs found in *Ager*^−/−^ mice. However, this was not the case, especially for the downregulated genes (in both neutrophils and lungs, *Figure 5C and F*), where many more downregulated genes are found uniquely in *Ager*^AHA/AHA^ mice than the ones that overlap with the downregulated genes in *Ager*^−/−^ mice. Based on the scope of the inflammatory genes that are differently regulated in *Ager*^−/−^ neutrophils (and most likely in other leukocytes as well), we believe caution will need to be taken when using *Ager*^−/−^ mice in inflammatory models. In many cases, the *Ager*^AHA/AHA^ mice might be a cleaner model for investigating the role of RAGE signaling.

The fact that most of the downregulated genes in *Ager*^AHA/AHA^ mice are not found in *Ager*^−/−^ mice is highly curious. Taking Gbp2b and Ifi44L as examples, the expression levels of both were completely abolished in neutrophils and lungs in *Ager*^AHA/AHA^ mice, but they maintained WT-like expression levels in *Ager*^−/−^ mice (*Figure 5—figure supplement 3*). If Gbp2b and Ifi44L (both are interferon γ responsive genes) were truly associated with RAGE activation, the only logical explanation would be that the activation of these genes was somehow preserved in *Ager*^−/−^ mice. Taking into considerations that many RAGE ligands can also cross-activate TLR4, it is plausible that in the absence of RAGE, certain RAGE ligands activate TLR4 instead, resulting in induction of Gbp2b and Ifi44L. In contrast, because RAGE^AHA^ mutant behaves essentially like a decoy receptor, being able to still bind the ligand without transducing signal, spillover of RAGE ligands to TLR4 would be unlikely. Indeed, LPS has been shown to be able to induce Gbp2b (synonymous with Gbp1) expression, which suggests that in theory activation of TLR4 with another ligand would also induce Gbp2b expression (*Kim et al., 2011*). Ongoing research in our lab seeks to establish the connection between RAGE signaling and Gbp2b-mediated defense signaling.

In summary, by using *Ager*^AHA/AHA^ mice, we presented strong evidence for an essential role of HS-RAGE interaction in bone remodeling and drug-induced liver injury. We further presented a new strategy in treating RAGE-associated diseases by blocking the HS-binding site of RAGE using mAb. Based on our RNA-seq study, we expect that HS-RAGE interaction would play essential roles in many

inflammatory responses that involve other types of leukocytes. The $Ager^{AHA/AHA}$ mice and the anti-RAGE mAb antibody we reported here would certainly be invaluable tools to further our understanding of the pathophysiological roles of RAGE. To evaluate the therapeutic potential of mAb B2, we plan to determine its efficacy in a number of inflammatory and tumor models and is in the process of humanization of B2 for testing in primates. A limitation of our study is that we only focused on a small portion of the HS-binding site located on C1 domain. It is perceivable that mAbs that target other epitopes within the HS-binding site would also effectively block RAGE signaling and we are hoping to identify more such mAbs to fully realize the therapeutic potential of RAGE antagonism.

# Materials and methods

**Key resources table**

| Reagent type (species) or resource | Designation | Source or reference | Identifiers | Additional information |
|---|---|---|---|---|
| Genetic reagent (*Mus musculus*) | $Ager^{AHA/AHA}$ | This paper | | Knockin mouse created by CRISPR |
| Genetic reagent (*M. musculus*) | $Ager^{-/-}$ | PMID:15173891 | MGI:2451038 | RAGE knockout mouse: Agertm1.1Arnd |
| Strain, strain background (*Escherichia coli*) | Origami-B (DE3) | MilliporeSigma | Cat#: 70837 | |
| Cell line (Hamster) | Flp-In-CHO Cell Line | Invitrogen | Cat#: R75807 RRID:CVCL_U424 | Directly purchased from manufacture, negative for mycoplasma |
| Cell line (*Homo-sapiens*) | MDA-MB-453 | ATCC | Cat#: HTB-131 RRID:CVCL_0418 | Identity authenticated by SRT profiling, negative for mycoplasma |
| Antibody | Anti-RAGE (rabbit monoclonal) | This paper | B2 | WB: 1 µg/ml Functional: 5–20 µg/ml |
| Antibody | Anti-RAGE (human monoclonal) | This paper | B2, rabbit-human chimeric | IHC: 1 µg/ml FC: 10 µg/ml In vivo: 10 mg/kg |
| Antibody | Anti-mouse Ly-6G-PE (rat monoclonal) | BD Biosciences | Cat#: 561104 RRID:AB_10563079 | FC: 1:100 |
| Antibody | Anti-mouse CD45-FITC (rat monoclonal) | BioLegend | Cat#: 103107 RRID:AB_312972 | FC: 1:100 |
| Antibody | Anti-mouse CD11b-APC (rat monoclonal) | BD Biosciences | Cat#: 561690 RRID:AB_10897015 | FC: 1:100 |
| Antibody | Anti-mouse Ly-6C-FITC (rat monoclonal) | BioLegend | Cat#: 128005 RRID:AB_1186134 | FC: 1:100 |
| Antibody | Anti-mouse CD3-PE (rat monoclonal) | BioLegend | Cat#: 100205 RRID:AB_312662 | FC: 1:100 |
| Antibody | Anti-mouse CD8a-APC (rat monoclonal) | BioLegend | Cat#: 100711 RRID:AB_312750 | FC: 1:100 |
| Antibody | Anti-mouse RAGE (rat monoclonal) | R&D Systems | Cat#: MAB1179 RRID:AB_2289349 | WB: 2 µg/ml |
| Antibody | Anti-HMGB1 (rabbit monoclonal) | Abcam | Cat#: Ab79823 RRID:AB_1603373 | WB: 1:10,000 |
| Recombinant DNA reagent | pcDNA5/FRT (plasmid) | Invitrogen | Cat#: V601020 | |
| Recombinant DNA reagent | pET21b (plasmid) | MilliporeSigma | Cat#: 69741 | |
| Recombinant protein | m-CSF | PeproTech | Cat#: 315-02 | |
| Recombinant protein | RANKL | PMID:15173891 | | |
| Recombinant protein | VC1 domain of mouse RAGE | PMID:23679870 | | |
| Sequence-based reagent | sgRNA | This paper | sgRNA | GGGGCCGCGT CTGGGGACTTGTG |
| Commercial assay or kit | Leukocyte Acid Phosphatase Kit | Sigma-Aldrich | Cat#: 387A-1KT | |

*Continued on next page*

*Continued*

| Reagent type (species) or resource | Designation | Source or reference | Identifiers | Additional information |
|---|---|---|---|---|
| Commercial assay or kit | RNeasy Mini Kit | QIAGEN | Cat#: 74134 | |
| Chemical compound, drug | Acetaminophen | Sigma-Aldrich | Cat#: A5000 | |
| Software, algorithm | ImageJ (v1.50i) | PMID:22930834 | RRID:SCR_003070 | https://imagej.nih.gov/ij/index.html |
| Software, algorithm | Graphpad Prism 7 | GraphPad Software | RRID:SCR_002798 | |
| Software, algorithm | AnalyzePro 1.0 | AnalyzeDirect Inc | RRID:SCR_005988 | https://analyzedirect.com |
| Software, algorithm | DESeq2 R package (v1.30.1) | PMID:25516281 | RRID:SCR_015687 | |
| Software, algorithm | clusterProfiler R package (v3.18.1) | PMID:22455463 | RRID:SCR_016884 | |

## Generation of *Ager*[AHA/AHA] knock-in mice

The knock-in project was performed at the Gene Targeting and Transgenic Shared Resource of Roswell Park comprehensive cancer center. Briefly, the synthesized sgRNA (which targets exon 6 of mouse *rage* gene as shown in *Figure 1*), repair template ssODN, and mRNA encodes for Cas9 were micro-injected into 1-cell stage fertilized egg of C57BL/6J strain, which were then surgically transferred into pseudopregnant foster mothers also of C57BL/6J background. Through CRIPSR-based homology-directed repair, the triple mutations were introduced into the *rage* locus. The pups were screened by next-generation sequencing, and the mice that bear correct mutations were mated to WT C57BL/6J mice to confirm germline transmission of the mutations. The F1 heterozygous mice were bred together to generate homozygous offspring. *Ager*[−/−] mice were originally gifted by A. Bierhaus (University of Heidelberg, Heidelberg, Germany).

## MicroCT analysis

Mouse femurs from different ages of WT, *Ager*[AHA/+], *Ager*[AHA/AHA], *Ager*[+/−], and *Ager*[−/−] mice were harvested, fixed for 48 hr in 10% neutral buffered formalin. A quantitative analysis of the gross bone morphology and microarchitecture was performed using ScanCo microCT 100 system (University at Buffalo). 3D reconstruction and bone microarchitecture analysis were performed using AnalyzePro (AnalyzeDirect Inc).

## TRAP staining

Mouse tibia from 10-week-old male mice were harvested, fixed for 48 hr in 10% neutral buffered formalin, and decalcified in 10% EDTA for 2 weeks at room temperature (RT). The samples were embedded in paraffin and sectioned at 5 µm for TRAP staining. The sections were counter stained with Fast Green.

## Osteoclastogenesis assay

Primary osteoblasts were isolated from calvaria of 5- to 8-day-old WT mice following an established protocol (*Taylor et al., 2014*). Osteoblasts ($5 \times 10^3$ cells/well) were seeded in a 96-well plate the day before starting the co-culture. Freshly isolated bone marrow cells (from WT, *Ager*[AHA/AHA], and *Ager*[−/−] mice) were suspended in 10 ml of α-MEM containing 10% fetal bovine serum and 1× penicillin/streptomycin, $10^{-7}$ M dexamethasone, and $10^{-8}$ M 1α- and 25-dihydroxyvitamin D3. About 100 µl of bone marrow cells were added to each well. The medium was replaced every 2 days thereafter until the appearance of giant osteoclasts. For B2 inhibition assay, 20 µg/ml B2-IgG or control IgG was included in the medium in selected wells. To visualize osteoclasts, the cells were fixed and stained for TRAP activity using a Leukocyte Acid Phosphatase Kit (Sigma-Aldrich). Osteoclasts were sub-grouped into four categories based on the number of nuclei and quantified to analyze the population of osteoclasts with different number of nuclei.

## Mineral apposition rate measurement

Eight-week-old male mice of WT, *Ager*[AHA/AHA], and *Ager*[−/−] were labeled at 9 days (20 mg/kg of calcein) and 2 days (30 mg/kg of Alizarin Red) prior to sacrifice via intraperitoneal injection. Tibias

were fixed with 10% formalin for 48 hr, treated with 5% KOH for 96 hr, and then processed and sectioned followed an established method (*Porter et al., 2017*). Images were taken with a Nikon Ci-S fluorescence microscope and merged using ImageJ software. Mineral apposition rate is the distance between the midpoints of the two labels (measured at multiple locations using ImageJ and averaged) divided by the time between the dye injections.

## Mouse model of APAP liver injury

WT, *Ager*^AHA/AHA^, and *Ager*^−/−^ mice were fasted overnight (12–15 hr) to deplete GSH stores before APAP (Sigma-Aldrich) administration. Fresh APAP was dissolved in warm (~50°C) sterile saline, cooled to 37°C, and injected intraperitoneally at 300 mg/kg. The plasma samples were collected 24 hr after APAP administrations for measuring ALT concentration. In general, the concentration of ALT reached its highest point at 24 hr after APAP overdose. For B2 protective treatment, WT mice were injected intraperitoneally with 10 mg/kg of B2-IgG 12 hr before APAP overdose (300 mg/kg). Plasma ALT was measured using the ALT Infinity reagent (Thermo Fisher Scientific) following the manufacturer's instructions. Livers were harvested, fixed for 24 hr in 10% neutral buffered formalin. Samples were further embedded in paraffin and sectioned at 4 µm for hematoxylin and eosin staining. The necrosis area was quantified with ImageJ.

## Neutrophil infiltration in liver necrosis

The identical liver lobe from different mice was collected, minced, and digested with 0.2% collagenase A for 1 hr at 37° with shaking, and the digested tissue was further dissociated into single-cell suspension by pipetting several times. Cells passed through 70-µm cell strainer were pelleted, resuspended in 1 ml phosphate-buffered saline (PBS) and overlayed onto 4 ml of 33% Percoll gradient. After centrifugation at RT for 15 min at 1400×*g*, cell pellet was collected as leukocytes which include polymorphonuclear (PMN) cells. Leukocytes were further treated with ACK Lysing Buffer for a short time to remove red blood cells and counted. The number of neutrophils was quantified by FACS after staining with rat anti-mouse Ly-6G-PE, Clone 1A8 (BioLegend), rat anti-mouse CD45-FITC (BioLegend), and rat anti-mouse CD11b-APC (BioLegend) for 15 min at 4°. Ly-6G high cells were counted as neutrophils.

## Air pouch model and neutrophil isolation

About 3 ml of sterile air was injected under the dorsal skin of WT, *Ager*^AHA/AHA^, and *Ager*^−/−^ mice. After 3 days, the pouch was refilled with 2 ml of sterile air. On the seventh day, 1 µg of LPS from *Escherichia coli* O111:B4 (Sigma-Aldrich) in 0.5% solution of carboxymethyl cellulose (sodium salt; Sigma-Aldrich) was injected into the air pouch. After 4 hr, mice were euthanized by isoflurane inhalation and the cells in the pouch were collected by washing with PBS. Cells were counted and stained with Ly-6G-PE to determine the percentage of PMNs.

## Leukocyte cell surface marker analysis

For analysis of PMNs, total cells collected from the air pouch were stained with Ly-6G-PE, Clone 1A8, rat anti-mouse CD45-FITC, and rat anti-mouse CD11b-APC, rat anti-mouse Ly-6C-PE, rat anti-mouse F4-80-PE for 15 min at 4°.

For analysis of T and B cells, splenocytes were stained with rat anti-mouse CD45-FITC, rat anti-mouse Ly-6C-PE, rat anti-mouse F4-80-PE, rat anti-mouse CD3-PE, rat anti-mouse CD4-APC, and rat anti-mouse CD8-APC. CD3^+^ cells were gated as T cells (including both CD4^+^ and CD8^+^ cells), and CD3^−^ cells were gated as B cells.

## RNA-seq transcriptome analysis

LPS-induced leucocytes (>95% PMNs) were collected from air pouch as described above and used for RNA extraction. For lung tissue samples, 30 mg thoroughly perfused lungs were homogenized by bullet blender 5E (Next Advance). Total RNA was prepared using RNeasy Mini Kit (QIAGEN) following the protocol. cDNA library construction and RNA sequencing were performed at Novogene Co. Inc (Sacramento, CA). RNA integrity and quantitation were assessed using the RNA Nano 6000 Assay Kit of the Bioanalyzer 2100 system (Agilent Technologies, CA, USA). A total amount of 1 µg RNA per sample was used as input material for the RNA sample preparations. Sequencing libraries were generated using NEBNext Ultra RNA Library Prep Kit for Illumina (NEB, USA) following the manufacturer's

recommendations. The clustering of the index-coded samples was performed on a cBot Cluster Generation System using PE Cluster Kit cBot-HS (Illumina) according to the manufacturer's instructions. After cluster generation, the library preparations were sequenced on an Illumina platform PE150 and paired-end reads (150 bp) were generated. Reference genome and gene model annotation files were downloaded from genome website browser (NCBI/UCSC/Ensembl, mm10) directly. Paired-end clean reads were aligned to the reference genome using Spliced Transcripts Alignment to a Reference (STAR) (*Dobin et al., 2013*).

DESeq2 R package (v1.30.1) was used for gene count normalization and differential expression analysis (*Love et al., 2014*). The list of $Ager^{AHA/AHA}$ and $Ager^{-/-}$ differentially expressed genes compared to WT was filtered by an absolute $\log_2$ fold change≥1 and a multiple testing corrected p value ≤0.05. This list of differentially expressed genes in $Ager^{AHA/AHA}$ and $Ager^{-/-}$ lungs or PMNs were further analyzed and annotated with GO analysis using clusterProfiler R package (v3.18.1) (*Yu et al., 2012*). A multiple testing corrected p value ≤0.05 cutoff was used for the applied conditional hypergeometric test for GO term overrepresentation of GO terms in molecular function and biological process ontologies. The used GO annotations were obtained from the Bioconductor *Mus musculus* annotation package. PMN and lung RNA-sequencing data have been deposited into the NCBI Gene Expression Omnibus database (accession number GSE174178).

## Development of B2

Rabbit monoclonal antibodies (mAbs) were raised against the murine RAGE V-C1 domain in its native conformation. The rabbit hybridomas were generated at Epitomics Inc (now part of Abcam). The hybridoma line 240 E-W3 was used to fuse with rabbit splenocytes. Out of the 4000 hybridoma clones that were screened, around 100 were able to detect mVC1 by direct ELISA. The supernatants of the positive clones were then screened for mAbs that specifically interacted with the HS-binding site. In this screening, instead of using WT mVC1, various mVC1 mutants (K43A-K44A, K39A-R104A, K107A, and R216A-R218A) were immobilized to ELISA plates. These mutants bear alanine mutations in the HS-binding residues. In this assay, mAbs were sought that show reduced binding to the mutants compared to the WT mVC1. In principle, the reduced binding would only occur if the residues that we mutated belong to the epitope recognized by the mAb. In this assay, the epitopes of B2 were found to include R216 and R218. Rabbit B2 was purified from hybridoma supernatant by Protein A chromatography.

## Direct ELISA for screening RAGE-binding mAbs

About 200 ng of human RAGE extracellular domain or R216A-R218A mutant were immobilized onto a 96-well high-binding ELISA plate. Plates were blocked with 5% bovine serum albumin (BSA) in PBS and incubated with 0.01–10 nM of rabbit B2 mAb for 1 hr at RT. Bound B2 mAbs were quantitated with anti-rabbit-HRP (Jackson ImmunoResearch) followed by the addition of HRP substrate (Thermo Fisher Scientific).

## Recombinant expression of B2

To produce B2 mAbs in large quantity for in vivo studies, we cloned the variable regions of B2 from hybridoma cells and grafted them onto human IgG1 constant region. The chimeric antibody was expressed in Chinese hamster ovary (CHO) cells. The pcDNA5_FRT plasmid purchased from Invitrogen was modified to insert the variable regions of B2 heavy and light chains in human IgG1 constant regions. Then the plasmid was transfected into CHO_Flp cells using lipofectamine 3000 Reagent (Invitrogen), and the positive clones were screened out over 10 days in the presence of Hygromycin B (50 mg/ml, Gibco). Cells with positive expression of the antibody were grown in CD CHO AGT expression medium (Thermo Fisher Scientific) with supplemental glutamine, D+ glucose, and Pen/Strep for 2 weeks. The media was then harvested, and the secreted antibody (IHH-IgG) was purified using a 5-ml HiTrap protein G column (Cytiva). The antibodies were buffer exchanged into PBS, and the concentration was determined using NanoDrop Spectrophotometer (Thermo Fisher Scientific). The purity of each batch was examined using SDS-PAGE. Purified antibodies were stored at 4°C before experiments. For B2-Fab, human chimeric B2 were digested with immobilized papain using Pierce Fab Preparation Kit (Thermo Fisher Scientific) following the instructions.

### Effect of B2 on RAGE binding on heparin

About 30 µg of mouse mVC1 was either directly loaded onto heparin Sepharose column, or loaded after 30 min incubation with 68 µg Fab fragment of B2 (1:1 molar ratio). B2-Fab bound mVC1 displayed greatly reduced binding to heparin column. The bound mVC1 was eluted with a salt gradient from 200 mM to 1.4 M NaCl, pH 7.1 in HEPES buffer.

### B2 cell surface binding by FACS

Breast tumor MDA-MB-453 was lifted from culture dish using Accutase (BioLegend) and incubated with human B2-IgG or B2-Fab in 100 µl PBS and 0.1% BSA for 45 min at 4°C. Bound B2-IgG was stained with goat anti-human IgG-Alexa 568 (1:1000; Invitrogen) for 30 min and analyzed by FACS. B2-Fab was stained with biotinylated goat anti-human IgG (1:1000, Invitrogen) followed by streptavidin-PE staining. In some experiments, cells were pretreated with recombinant heparin lyases III (5 milliunits/ml) for 15 min at RT prior to binding experiments. The MDA-MB-453 cell line was obtained from ATCC (HTB-131). Its identity was authenticated by SRT profiling, and it was tested negative for mycoplasma in the lab.

### Immunohistochemistry

Lungs from WT and *Ager*$^{-/-}$ mice were perfused, harvested, and fixed for 24 hr in 10% neutral buffered formalin. Samples were further embedded in paraffin and sectioned at 5 µm for RAGE staining with 1 µg/ml human B2-IgG. After staining with biotinylated rabbit anti-human IgG secondary antibody (1:200), the sections were developed using the ABC system (Vector Laboratories) and the cell nuclei were counter stained with 15% Ehrlich's hematoxylin (Electron Microscopy Sciences).

### Effect of antibodies on RAGE-ligand binding

HMGB1 or S100b (200 ng) was immobilized and the plate was blocked with 5% BSA. Biotinylated-mouse B2 V-C1 domain (200 ng/ml) was pre-incubated with rabbit IgG, polyclonal or monoclonal anti-RAGE (all at 5 µg/ml) for 30 min at RT before being added to the plate. The percentage of antibody inhibition was calculated by comparing the absorbance obtained in the presence of specific antibodies to the absorbance obtained in the presence of control rabbit IgG (which was set to 100%).

### Expression of recombinant murine and human RAGE

WT mouse RAGE V-C1 domains (mVC1-WT) and human sRAGE (V-C1-C2 domains) were produced in *E. coli* cells. Purification was carried out using HiTrap SP cation exchange column at pH 7.8, followed by SEC on a Superdex 200 column (GE Healthcare). The recombinant protein was purified to >98% pure as examined by silver staining. Mouse RAGE triple mutant R216A-R217H-R218A (mVC1-AHA) was confirmed by sequencing and purified as described above for WT protein.

### Heparin-Sepharose chromatography and analytical size-exclusion chromatography

To characterize the binding of mVC1 and mutant to heparin, 100 µg of purified mVC1-WT or mVC1-AHA were applied to a HiTrap heparin-Sepharose column and eluted with a salt gradient from 200 mM to 1.4 M NaCl, pH 7.1 (HEPES buffer). The conductivity measurements at the peak of the elution were converted to the concentration of NaCl based on a standard curve. For analyses of mVC1 and HS dodecasaccharide (H12) complex, purified mVC1-WT or mVC1-AHA (40 µg) was incubated with HS dodecasaccharide (H12, 4 µg) in 20 mM Tris, 150 mM NaCl, pH 7.4, at 4°C overnight. H12 is chemoenzymatically synthesized HS 12mer with NS, 2S, and 6S modification (gift from Dr. Jian Liu, UNC Chapel Hill). All complexes were resolved on a Superdex 200 (10/300 mm) gel filtration column using 20 mM Tris, 150 mM NaCl, pH 7.4, at 4°C.

### HMGB1 binding ELISA

Binding affinity of mRAGE-WT and mRAGE-AHA to HMGB1 was measured by ELISA. Briefly, 96-well plate was coated by 200 ng of mRAGE-WT and mRAGE-AHA protein and blocked by 1% BSA. Biotinylated HMGB1 with concentrations from 50 ng/ml to 1000 ng/ml was added into wells and incubated for 2 hr followed by incubation with streptavidin-HRP for 30 min. About 50 µl of HRP substrate solution was added for developing, and the reaction was stopped by adding 50 µl of 1 M $H_2SO_4$. The

absorbance at 450 nm was measured by a plate reader. Apparent $K_d$ value was calculated using Prism software.

## Western blot analysis

One lung lobe from WT, *Ager*$^{AHA/AHA}$, or *Ager*$^{-/-}$ mice was collected and homogenized/lysed in lysis buffer (50 mM Tris-HCl, pH 7.4, 1% Triton X-100). After clearing by centrifugation, the lysate was boiled in SDS sample with 5% β-mercaptoethanol and analyzed by gel electrophoresis using SureGel 4%–20% Bis-Tris gel (Genscripts). After transfer, the PVDF membrane was probed with a rat anti-RAGE mAb (R&D Systems), rabbit anti-HMGB1 mAb (Abcam), or our rabbit B2.

## Statistical analysis

All data are expressed as means ± SEM. Statistical significance of differences between experimental and control groups was analyzed by two-tailed unpaired Student's t-test, between multiple groups by one-way analysis of variance (ANOVA) followed by Dunnett's or Tukey's multiple comparisons test using Prism software (version 7.03; GraphPad Software Inc).

## Acknowledgements

The authors thank the Optical Imaging and Analysis Facility of School of Dental Medicine, University at Buffalo, for assistance with μCT analysis. The authors also thank the Gene Targeting and Transgenic Shared Resource of Roswell Park Comprehensive Cancer Center for help with generating RAGE knock-in mice. The authors further thank Dr. Jian Liu (UNC Chapel Hill) for gifting us HS 12mer oligosaccharide. This work is supported by the National Institutes of Health Grants R01AR07017 and R01HL094463 (to DX); R01GM125095 (to EPS); R01GM114179, R21AI138195 and R01CA246785 (to DKS); and Buffalo Accelerator Funds (to DX).

## Additional information

### Competing interests

Miaomiao Li, Ding Xu: is one of the inventors for an international patent (pending, WO 2021/087462) that covers the sequence and use of anti-RAGE mAb B2. The other authors declare that no competing interests exist.

### Funding

| Funder | Grant reference number | Author |
| --- | --- | --- |
| National Institute of Arthritis and Musculoskeletal and Skin Diseases | R01AR07017 | Ding Xu |
| National Heart, Lung, and Blood Institute | R01HL094463 | Ding Xu |
| Buffalo Accelerator Funds | | Ding Xu |
| National Institute of General Medical Sciences | R01GM125095 | Eric P Schmidt |
| National Institute of General Medical Sciences | R01GM114179 | Dhaval K Shah |
| National Institute of Allergy and Infectious Diseases | R21AI138195 | Dhaval K Shah |
| National Cancer Institute | R01CA246785 | Dhaval K Shah |

The funders had no role in study design, data collection and interpretation, or the decision to submit the work for publication.

## Author contributions
Miaomiao Li, Conceptualization, Formal analysis, Investigation, Methodology, Writing – original draft, Writing – review and editing; Chih Yean Ong, Lisi Tan, Ashwni Verma, Formal analysis, Investigation, Methodology; Christophe J Langouët-Astrié, Formal analysis, Investigation, Methodology, Writing – original draft, Writing – review and editing; Yimu Yang, Xiaoxiao Zhang, Investigation, Methodology; Dhaval K Shah, Conceptualization, Funding acquisition, Resources, Supervision, Writing – review and editing; Eric P Schmidt, Conceptualization, Formal analysis, Funding acquisition, Resources, Supervision, Writing – original draft, Writing – review and editing; Ding Xu, Conceptualization, Data curation, Formal analysis, Funding acquisition, Investigation, Methodology, Project administration, Resources, Supervision, Validation, Writing – original draft, Writing – review and editing

## Author ORCIDs
Ashwni Verma (iD) http://orcid.org/0000-0003-3717-0233
Xiaoxiao Zhang (iD) http://orcid.org/0000-0003-1321-0798
Ding Xu (iD) http://orcid.org/0000-0001-9380-2712

## Ethics
All animal works in this study have been approved by the institutional animal care and use committee of the University at Buffalo (protocol number: ORB14126N and ORB18018).

## Decision letter and Author response
Decision letter https://doi.org/10.7554/eLife.71403.sa1
Author response https://doi.org/10.7554/eLife.71403.sa2

---

## Additional files

### Supplementary files
• Supplementary file 1. List of differentially expressed genes (DEGs) from WT, $Ager^{AHA/AHA}$ and $Ager^{-/-}$ PMNs.

• Supplementary file 2. Gene ontology analysis of genes from $Ager^{-/-}$ PMNs.

• Supplementary file 3. Gene ontology analysis of genes from $Ager^{AHA/AHA}$ PMNs.

• Supplementary file 4. List of differentially expressed genes (DEGs) from WT, $Ager^{AHA/AHA}$ and $Ager^{-/-}$ lungs.

• Supplementary file 5. Gene ontology analysis of genes from WT, $Ager^{AHA/AHA}$ and $Ager^{-/-}$ lungs.

• Transparent reporting form

### Data availability
PMN and lung RNA-sequencing data have been deposited into the NCBI Gene Expression Omnibus database (accession number GSE174178). All data generated or analysed during this study are included in the manuscript and supporting files. Source data files have been provided for all the figures and figure supplements.

The following dataset was generated:

| Author(s) | Year | Dataset title | Dataset URL | Database and Identifier |
|---|---|---|---|---|
| Li M, Langouët-Astrié CJ, Xu D | 2021 | mRNA-seq analysis of RAGE knock-in (RageAHA/AHA) and RAGE-/- mice | http://www.ncbi.nlm.nih.gov/geo/query/acc.cgi?acc=GSE174178 | NCBI Gene Expression Omnibus, GSE174178 |

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
