## [Editor Report]

The Receptor for Advanced Glycation End-products (or RAGE) has garnered great interest over the past 20 years for its role in the complications of diabetes mellitus and in Alzheimer's disease, atherosclerosis and other inflammatory disorders. RAGE has several ligands. This paper explores the role of heparan sulfate in the oligomerization of RAGE and the role of oligomerization in vivo function using mouse knockout models. The authors report that knock-in mice, with RAGE is mutated at sites responsible for heparan sulfate binding and oligomerization, phenocopy RAGE knockout mice. They further validate the idea that this knock-in mouse, which preserves expression and binding of RAGE, may be a valuable model for studying this important molecule in disease pathogenesis.

---

## [Decision Letter]

**Decision letter after peer review:**

Thank you for submitting your article "Heparan Sulfate-dependent RAGE oligomerization is indispensable for pathophysiological functions of RAGE" for consideration by *eLife*. Your article has been reviewed by 3 peer reviewers, and the evaluation has been overseen by a Reviewing Editor and Paul Noble as the Senior Editor. The following individual involved in review of your submission has agreed to reveal their identity: Lianchun Wang (Reviewer #3).

Essential revisions:

The Reviewing Editor feels that the following comments by the 3 reviewers are essential ones that should be addressed to have the paper accepted. If the authors can straightforwardly address the other concerns, this will be great, but please try hard to address the following in your rebuttal. The Editors are committed to getting the paper published without further unneeded delays.

From the Reviewing Editor, please be sure to address the following if possible:

Reviewer #1: Comment #1 – please do your best to address this as you can without the need for many additional experiments.

Reviewer #2: Can you try to address Comments #2 and #4. These seem reasonable. If you do not have complete bone formation data, then explain that in your reply back.

Reviewer #3: Do you have any data to address Comment #9? This would be good to do. Can you address Comment #4, which seems straightforward?*Reviewer #1:*

In this manuscript, Li et al. examined the physiological significance of the interaction of RAGE, a transmembrane inflammatory receptor, with the cell surface polysaccharide heparan sulfate (HS) in vivo. The authors started by producing a triple mutant of RAGE with point mutations in the HS binding site. This mutant protein displayed normal binding to a known RAGE ligand (HMGB1) but reduced binding to heparin-Sepharose and impaired oligomerization when incubated with an HS oligosaccharide. Subsequently, they generated a novel CRISPR knock-in mouse model containing the same point mutations in RAGE (RageAHA/AHA) to specifically disrupt its interaction with HS in vivo. Strikingly, their RAGE knock-in mouse showed defects in bone remodeling and response to liver injury similar to the RAGE knockout mouse. They show that RageAHA/AHA mice display increased bone mass and volume compared to wild-type mice (≥10 weeks of age), and RAGE gene dosage studies showed the dominant negative impact of RAGE-AHA on bone remodeling. In addition, the RAGE-AHA mutation caused a reduction in osteoclastogenesis in vivo and in vitro. In a model of neutrophil-mediated injury using a sublethal dose of acetaminophen, RageAHA/AHA mice displayed a reduction in neutrophil infiltration and liver damage markers similar to the RAGE knockout model. RNA-sequencing of isolated neutrophils and lung tissue from all three genotypes revealed large disparities in differential gene expression between the Rage-/- and RageAHA/AHA mice, suggesting that specifically disrupting RAGE-HS interactions may be a cleaner model for investigating the physiological role of RAGE signaling. To follow up on their findings, the authors generated a monoclonal antibody that specifically targets the HS-binding site of RAGE and demonstrated that pharmacological targeting of RAGE-HS interactions phenocopies the RAGE knockout mouse strain and decreases liver injury and modulates bone remodeling.

Overall, this study nicely follows up on the authors' previous work and reveals the essential role and physiological significance of HS-RAGE oligomerization, which impacts RAGE signaling in vivo. The mouse model and antibody developed here are the main strengths of this study, which provides new innovative tools for studying and pharmacologically targeting RAGE signaling. Importantly, this work presents a new strategy to inhibit RAGE signaling without affecting other RAGE-ligand interactions, which may have direct application to helping patients dealing with relevant inflammatory disorders.

Although the work is well done overall, to support the conclusions made in this study, the following issues would need to be addressed:

(1) It is important to characterize the impact that the introduced point mutations have on RAGE protein stability and folding, which can affect protein function. This is particularly important since the authors compare their RAGEAHA mouse model to RAGE knockout mice in multiple assays and biological systems. While western blot data are included showing equal protein expression in WT and RageAHA/AHA mice, this does not address protein function and stability. Techniques such as differential scanning calorimetry (DSC) and/or CD spectroscopy should be used to compare WT and RAGEAHA protein stability/folding for the RAGE VC1 proteins.

(2) In previous work (Xu et al. 2013), the authors showed that an HS 16-mer is unable to induce oligomerization of the RAGE R216A-R218A mutant (Suppl. Information). In the current study, a 12-mer (H12) is used and seems to partially induce oligomerization (Figure 1C). Can the authors comment on this discrepancy in chain length and oligomerization? Also, can the authors include the structure and sulfation pattern of the HS dodecasaccharides used in these experiments? Is this a mixture of heparin-derived polysaccharides? None of this information was located in the materials/methods provided.*Reviewer #2:*

It is common that many receptors bind to heparan sulfate. Thus, It is important to determine the functional significance of heparan sulfate's interaction with receptors. In this paper, the authors used two important tools (RAGE knock-in mice, RageAHA/AHA and blocking HS-RAGE antibody) to address this issue and found that RageAHA/AHA mice (point mutations to disrupt HS-RAGE interaction) phenocopied Rage-/- mice in deficits of bone remodeling and neutrophil-mediated liver injury. The study does not produce novel insights into RAGE's function or into the molecular and cellular mechanisms that underlie RAGE's functions. There are many cellular questions that remain unanswered in the manuscript that will need addressing in future work.

The authors used two important tools (RAGE knock-in mice, RageAHA/AHA and blocking HS-RAGE antibody) to investigate the functions of RAGE binding to heparan sulfate.

They provide evidence for important functions of the heparan sulfate-RAGE interaction in regulating osteoclast development and bone remodeling, and in neutrophil-mediated liver injury. These are important findings, but the concerns described below should be addressed.

1. This study provides evidence that RageAHA/AHA mice (point mutations to disrupt HS-RAGE interaction) phenocopied Rage-/- mice in deficits of bone remodeling and neutrophil-mediated liver injury. However, these studies do not reveal new insights into RAGE's function nor molecular and cellular mechanisms that underlie RAGE's functions.

2. Many questions remain unanswered. For example, does RageAHA/AHA affect its surface or subcellular distribution? Does RageAHA/AHA impair RANKL and/or integrin signaling pathways in osteoclast development?

3. Please verify the RageAHA/AHA's deficit in RAGE oligomerization in primary cultured cells.

4. For bone remodeling or osteoclast deficit, it is necessary to examine both bone formation and bone resorptions in vivo and in vitro.

5. For the blocking antibody treatments in vivo, does it affect bone remodeling?

6. To prove the blocking antibody's specificity in vivo, it is better to include Rage-/- mice for the treatments as a control.

*Reviewer #3:*

Receptor for advanced glycation endproducts (RAGE) involves many disease conditions and has attracted tremendous interest to understand its biological functions and related regulation. RAGE functions as an oligomer on the cell surface. Extending from their previous work showing heparan sulfate (HS) functions to maintain RAGE oligomerization on the endothelial cell surface in vitro, this research group has directed further work toward understanding the physiological significance of the HS-mediated RAGE oligomerization in vivo. They have generated and examined novel RAGE knock-in (RageAHA/AHA) mice, in which point mutations were introduced to specifically disrupt HS-RAGE interaction. The introduced mutations do not affect ligand binding, but specifically, disrupt HS-RAGE interaction and HS-mediated RAGE oligomerization in vitro. The RageAHA/AHA mice appear normal but develop an osteopetrotic phenotype associated with impaired osteoclastogenesis and are protected from neutrophil-mediated liver injury induced by APAP overdose, phenocopying the Rage-/- mice. The authors also generated a monoclonal antibody B2 that targets the HS-binding site of RAGE. B2 blocked RAGE-dependent osteoclastogenesis and APAP-induced liver injury, phenocopying the RageAHA/AHA mice. The RNAseq analysis of neutrophils and lungs determined the changes of the transcriptome in RageAHA/AHA mice were much more restricted than the Rage-/- mice. These results clearly show that HS-induced RAGE oligomerization is essential for RAGE signaling and suggest that targeting the HS-RAGE interaction as an alternative strategy to antagonize RAGE. However, the RNAseq experiments did not reveal any functional alteration associated particularly with AHA mutations in mice. In addition, some experiments may need additional controls to better support the major conclusions in the manuscript. In general, the findings of this study are novel and may impact multiple related fields. The manuscript is well written.

1. mVC1-AHA still has 5 basic residues in the V domain involved in the HS interaction. As shown in Figure 1B, mVC1-AHA still binds to HS, although at a lower binding than mVC1-WT. It will be important to determine their relative binding affinity to HS or heparin. The remaining HS-binding-related residues may be a partial reason for the minor phenotype difference between RageAHA/AHA and Rage-/- mice. Such new data might strengthen the Discussion too.

2. Figure 1A cartoon indicates HS stabilizes RAGE dimer. Figure 1C data appear to suggest that HS is not essential for dimer formation, instead of hexamer. Is that right?

3. CRISPR gene manipulation, even using single-stranded donor oligonucleotide, might introduce out-off target mutations resulting in a phenotype unrelated to RAGE. This might be one potential cause of the phenotype difference between the RageAHA/AHA and the Rage-/- mice. in vitro rescue experiments with osteoclastogenesis and neutrophil function assay will help to exclude this possibility.

4. The phenotype displayed in the RageAHA/AHA mice are more related to osteoclasts and neutrophils, it will be more relevant to check RAGE expression level in these two cell populations in addition to whole lung tissue (Figure 1F).

5. Figure 1D detected the binding of mVC1-AHA to HMGB1, how about other ligands? It would be important to have a second method, such as SPR, to confirm the induced mutations only affect RAGE`s interaction with HS, not its ligands. Furthermore, how about the ligand-RAGE binding in the presence of HS or heparin – expecting HS/heparin will not affect the binding of mVC1-AHA to its ligands?

6. In the RAGE gene dosage study, in the RageAHA/+ cells only 25% RAGE forms functional dimers with intact HS-binding site, and in the Rage+/- cells, 100% RAGE forms functional dimers. It is important to determine if the Rage+/- cells indeed express 50% reduced cell surface RAGE. This data will strengthen the discussion of the levels of functional RAGE dimer formation.

7. In Figure 6A, antibody B2 showed a binding affinity of 4 nM, appearing to have a good binding affinity to RAGE. If this is true, B2 might stain positively for RageAHA/AHA lung (this needs to be included in Figure 6E) and a positive band in WB (Figure 6F). In addition, Figure 6F is missing the loading control.

8. Figure 7. B2 inhibits RAGE-dependent biological processes in the cell and animal models without including Rage-/- and RageAHA/AHA mice as controls. These controls are required to support the conclusion that antibody B2 specifically disrupts HS-RAGE interaction.

9. It will be helpful to determine if RAGE ligand levels are altered in Rage-/- and RageAHA/AHA mice. This is mentioned in the discussion without any experimental data. Including these data may help to interpret the observed phenotypes, especially the RNAseq analysis which did not find out the major mechanism underlying the observed phenotypes and the authors' claim the RageAHA/AHA mouse might represent a cleaner genetic model to study physiological roles of RAGE in vivo compared to Rage-/- mice.

---

## [Author Response]

Reviewer #1:[…] (1) It is important to characterize the impact that the introduced point mutations have on RAGE protein stability and folding, which can affect protein function. This is particularly important since the authors compare their RAGEAHA mouse model to RAGE knockout mice in multiple assays and biological systems. While western blot data are included showing equal protein expression in WT and RageAHA/AHA mice, this does not address protein function and stability. Techniques such as differential scanning calorimetry (DSC) and/or CD spectroscopy should be used to compare WT and RAGEAHA protein stability/folding for the RAGE VC1 proteins.

In Figure 1, we characterized the binding affinity of RAGE-AHA mutant to HMGB1 and showed that the binding is identical to WT RAGE. We see this as the most definitive demonstration of structural integrity of the mutant. In previously publication, we also examined human R216A-R218A mutant and showed its binding to HMGB1 and S100b were normal (Xu et al. 2013).

2) In previous work (Xu et al. 2013), the authors showed that an HS 16-mer is unable to induce oligomerization of the RAGE R216A-R218A mutant (Suppl. Information). In the current study, a 12-mer (H12) is used and seems to partially induce oligomerization (Figure 1C). Can the authors comment on this discrepancy in chain length and oligomerization? Also, can the authors include the structure and sulfation pattern of the HS dodecasaccharides used in these experiments? Is this a mixture of heparin-derived polysaccharides? None of this information was located in the materials/methods provided.

12mer is the minimum length that is required for binding to RAGE and inducing RAGE hexamerization. In the 2013 paper we used a human sRAGE (VC1C2 domains) R216A-R218A, while in this study we used mouse VC1 domains. In both cases, we observed almost complete loss of hexamer, but as the reviewer observed, the mouse VC1 R216A-R217H-R218A did maintain some ability to form dimers and likely exist in an equilibrium between monomer and dimer. The differences in their tendency to form dimer in the presence of HS oligosaccharide could be due to two reasons. First, the species difference might contribute to how easy the dimer can be formed. Secondly, mouse VC1, being smaller than human sRAGE (VC1C2), might be a little easier to form dimer.

We use purified size-defined heparin-derived oligosaccharides in Xu 2013 paper, and chemoenzymatically synthesized heparan sulfate 12mer in the current study, the structure information is included in the material part now.

Reviewer #2:[…] The authors used two important tools (RAGE knock-in mice, RageAHA/AHA and blocking HS-RAGE antibody) to investigate the functions of RAGE binding to heparan sulfate.They provide evidence for important functions of the heparan sulfate-RAGE interaction in regulating osteoclast development and bone remodeling, and in neutrophil-mediated liver injury. These are important findings, but the concerns described below should be addressed.1. This study provides evidence that RageAHA/AHA mice (point mutations to disrupt HS-RAGE interaction) phenocopied Rage-/- mice in deficits of bone remodeling and neutrophil-mediated liver injury. However, these studies do not reveal new insights into RAGE's function nor molecular and cellular mechanisms that underlie RAGE's functions.

We believe the mechanisms revealed in this study certainly provide new insights on how RAGE signaling is initiated as the cells surface, and what takes to assemble a functional RAGE receptor complex. These are all critical molecular and cellular mechanisms that underlie RAGE’s functions.

2. Many questions remain unanswered. For example, does RageAHA/AHA affect its surface or subcellular distribution? Does RageAHA/AHA impair RANKL and/or integrin signaling pathways in osteoclast development?

One way we can assess whether RAGE-AHA mutant altered cell surface localization is to examine the staining pattern of RAGE on lung tissue sections. When staining RAGE-AHA lung with B2, we normally have very weak or no staining compared to WT lung. But if we use B2 at higher concentration, we can observe staining in the same cell surface staining pattern as the WT lung (as shown in Author response image 1). This data shows that the surface localization of RAGE-AHA mutant was not altered.

**Author response image 1. sa2fig1:** 

Previous study by Dr. Xiong group has shown clearly that impaired RAGE signaling in osteoclasts led to reduction in RANKL and integrin signaling pathways. Since RAGE-AHA mice display the same osteoclasts phenotype as the RAGE-/- mice, we would logically conclude that in RAGE-AHA cells the same signaling pathways were affected.

3. Please verify the RageAHA/AHA's deficit in RAGE oligomerization in primary cultured cells.

Unfortunately, there is no straightforward way to directly assess RAGE oligomerization in any cells. This would require an oligomeric RAGE specific antibody that can distinguish monomeric RAGE from RAGE oligomer. Such antibody is not available.

4. For bone remodeling or osteoclast deficit, it is necessary to examine both bone formation and bone resorptions in vivo and in vitro.

We actually performed in vivo bone formation analysis comparing WT, RAGE-AHA and RAGE-/- mice. No difference was detected in RAGE-AHA and RAGE-/-, which suggests that bone formation was largely normal in both mice. The data was included as Supplemental Figure S3.

5. For the blocking antibody treatments in vivo, does it affect bone remodeling?

In our in vivo experiment injecting B2, the duration of the experiment was only two days. We don’t expect such a short treatment would cause any change in bone remodeling. We did plan to assess the effect of B2 treating on bone remodeling, but this would require much longer treatment (>4 weeks), higher dose, and likely targeted delivery method to bone.

6. To prove the blocking antibody's specificity in vivo, it is better to include Rage-/- mice for the treatments as a control.

We have demonstrated the specificity of B2 by using lung tissue sections from RAGE-/- mice. This is commonly used standard in pharmaceutical industry to show the specificity of an antibody. As shown in Figure 6E, B2 gave absolutely no staining for RAGE-/- lung. Based on the fact the B2 treatment generated a similar phenotype as same in RAGE-AHA mice, we believe that the whole phenotype is due to unspecific effect of B2 is slim to none.

Reviewer #3:[…] 1. mVC1-AHA still has 5 basic residues in the V domain involved in the HS interaction. As shown in Figure 1B, mVC1-AHA still binds to HS, although at a lower binding than mVC1-WT. It will be important to determine their relative binding affinity to HS or heparin. The remaining HS-binding-related residues may be a partial reason for the minor phenotype difference between RageAHA/AHA and Rage-/- mice. Such new data might strengthen the Discussion too.

We agree that the remaining binding capacity of RAGE-AHA might contribute the minor phenotypic differences between RAGE-AHA and RAGE-/- mice. However, the key defect of RAGE-AHA mutant is that it couldn’t form stable hexamer anymore (Figure 1C), which is required for normal signaling. Therefore, we believe that knowing the exact binding affinity of RAGE-AHA to heparin would not change how we understand the mechanism in any significant way.

2. Figure 1A cartoon indicates HS stabilizes RAGE dimer. Figure 1C data appear to suggest that HS is not essential for dimer formation, instead of hexamer. Is that right ?

It’s important to remember that RAGE-AHA still retain partial binding capability to HS. Therefore in the presence of HS oligosaccharide, RAGE-AHA can still bind HS (albeit weaker) and form dimer. But the key is that the dimer is not stable, as reflected in Figure 1C, where the RAGE-AHA/H12 complex showed a much broader peak and overlaps with the monomer peak. This SEC profile suggests that in the presence of HS, RAGE-AHA exists in an equilibrium between monomer and dimer, which likely prevented it to form stable hexamer.

3. CRISPR gene manipulation, even using single-stranded donor oligonucleotide, might introduce out-off target mutations resulting in a phenotype unrelated to RAGE. This might be one potential cause of the phenotype difference between the RageAHA/AHA and the Rage-/- mice. in vitro rescue experiments with osteoclastogenesis and neutrophil function assay will help to exclude this possibility.

We agree that off target mutations might be introduced during CRISPR gene manipulation, but the chances are extremely slim in our case for two reasons. First, the transgenic mice were back crossed to WT BL/6 for three generations, which would greatly dilute any off-target mutations. Secondly, we have sequenced 6 potential off targets sites in RAGE AHA/AHA mice and found that all sites were completely normal.

4. The phenotype displayed in the RageAHA/AHA mice are more related to osteoclasts and neutrophils, it will be more relevant to check RAGE expression level in these two cell populations in addition to whole lung tissue (Figure 1F).

We appreciate the reviewer’s comment and have analyzed RAGE expression in neutrophils and osteoclasts by WB. However, because these cells express RAGE at much lower level compared to lung cells, none of the anti-RAGE antibodies we tried could detect distinct bands of RAGE from these cell lysates. We have tested B2 mAb, a rat anti-mouse RAGE (R&D Systems MAB1179), a goat anti-mouse RAGE (R&D System AF1179) and another mouse anti-RAGE mAb we recently developed. Of note, all four antibodies could detect specific bands of RAGE from less than 2 µg of lung lysate.

5. Figure 1D detected the binding of mVC1-AHA to HMGB1, how about other ligands? It would be important to have a second method, such as SPR, to confirm the induced mutations only affect RAGE`s interaction with HS, not its ligands. Furthermore, how about the ligand-RAGE binding in the presence of HS or heparin? – expecting HS/heparin will not affect the binding of mVC1-AHA to its ligands.

In Xu 2013 paper, we examined binding of human sRAGE R216A-R218A to both HMGB1 and S100b. We have unpublished data showing the ligand-RAGE binding was not affected in the presence of heparin.

6. In the RAGE gene dosage study, in the RageAHA/+ cells only 25% RAGE forms functional dimers with intact HS-binding site, and in the Rage+/- cells, 100% RAGE forms functional dimers. It is important to determine if the Rage+/- cells indeed express 50% reduced cell surface RAGE. This data will strengthen the discussion of the levels of functional RAGE dimer formation.

We did observe a bone phenotype of RAGE+/- mice. The only logical explanation would be that they have reduced RAGE expression, which led to a phenotype. Knowing whether they indeed express 50% of normal level would not change our conclusion in any meaningful way.

7. In Figure 6A, antibody B2 showed a binding affinity of 4 nM, appearing to have a good binding affinity to RAGE. If this is true, B2 might stain positively for RageAHA/AHA lung (this needs to be included in Figure 6E) and a positive band in WB (Figure 6F). In addition, Figure 6F is missing the loading control.

In tissue staining, we did observe faint RAGE-AHA/AHA staining as the reviewer predicted. This data is now included in Figure 6E. For WB, we did not observe any band of RAGE-AHA even after extended exposure. The loading control of Figure 6F was actually shown in Figure 1F.

8. Figure 7. B2 inhibits RAGE-dependent biological processes in the cell and animal models without including Rage-/- and RageAHA/AHA mice as controls. These controls are required to support the conclusion that antibody B2 specifically disrupts HS-RAGE interaction.

We have shown that B2 specifically disrupts HS-RAGE interaction in Figure 6B. We further demonstrated the specificity of B2 by using lung tissue sections from RAGE-/- mice, which is a commonly used standard in pharmaceutical industry to show the specificity of an antibody. Therefore we believe that the whole phenotype is due to unspecific effect of B2 is slim to none.

9. It will be helpful to determine if RAGE ligand levels are altered in Rage-/- and RageAHA/AHA mice. This is mentioned in the discussion without any experimental data. Including these data may help to interpret the observed phenotypes, especially the RNAseq analysis which did not find out the major mechanism underlying the observed phenotypes and the authors' claim the RageAHA/AHA mouse might represent a cleaner genetic model to study physiological roles of RAGE in vivo compared to Rage-/- mice.

We have now included in supplemental Figure S7 the analysis of the highly expressed RAGE ligands in neutrophils and lung. We found that expression of all RAGE ligands in RAGE-AHA mice were unaltered compared to WT mice. Expression of a few RAGE ligands were reduced moderately in RAGE-/- mice.